# D2C: Diffusion-Decoding Models for Few-Shot Conditional Generation

**Abhishek Sinha**[*]
Department of Computer Science
Stanford University
a7b23@stanford.edu

**Jiaming Song**[*]
Department of Computer Science
Stanford University
tsong@cs.stanford.edu

**Chenlin Meng**
Department of Computer Science
Stanford University
chenlin@cs.stanford.edu

**Stefano Ermon**
Department of Computer Science
Stanford University
ermon@cs.stanford.edu

## Abstract

Conditional generative models of high-dimensional images have many applications, but supervision signals from conditions to images can be expensive to acquire. This paper describes Diffusion-Decoding models with Contrastive representations (D2C), a paradigm for training unconditional variational autoencoders (VAEs) for few-shot conditional image generation. D2C uses a learned diffusion-based prior over the latent representations to improve generation and contrastive self-supervised learning to improve representation quality. D2C can adapt to novel generation tasks conditioned on labels or manipulation constraints, by learning from as few as 100 labeled examples. On conditional generation from new labels, D2C achieves superior performance over state-of-the-art VAEs and diffusion models. On conditional image manipulation, D2C generations are two orders of magnitude faster to produce over StyleGAN2 ones and are preferred by $50\% - 60\%$ of the human evaluators in a double-blind study. We release our code at https://github.com/jiamings/d2c.

## 1 Introduction

Generative models trained on large amounts of unlabeled data have achieved great success in various domains including images [8, 50, 75, 42], text [56, 2], audio [26, 71, 93, 62], and graphs [36, 67]. However, downstream applications of generative models are often based on various conditioning signals, such as labels [61], text descriptions [60], reward values [101], or similarity with existing data [45]. While it is possible to directly train conditional models, this often requires large amounts of paired data [57, 74] that are costly to acquire. Hence, it would be desirable to learn conditional generative models using large amounts of unlabeled data and as little paired data as possible.

Contrastive self-supervised learning (SSL) methods can greatly reduce the need for labeled data in discriminative tasks by learning effective representations from unlabeled data [95, 37, 35], and have also been shown to improve few-shot learning [39]. It is therefore natural to ask if they can also be used to improve few-shot generation. Latent variable generative models (LVGM) are a natural candidate for this, since they already involve a low-dimensional, structured latent representation of the data they generate. However, generative adversarial networks (GANs, [34, 50]) and diffusion

---

[*]Equal contribution.

models [42, 84], lack explicit tractable functions to map inputs to representations, making it difficult to optimize latent variables with SSL. Variational autoencoders (VAEs, [52, 77]), on the other hand, can naturally adopt SSL through their encoder model, but they typically have worse sample quality.

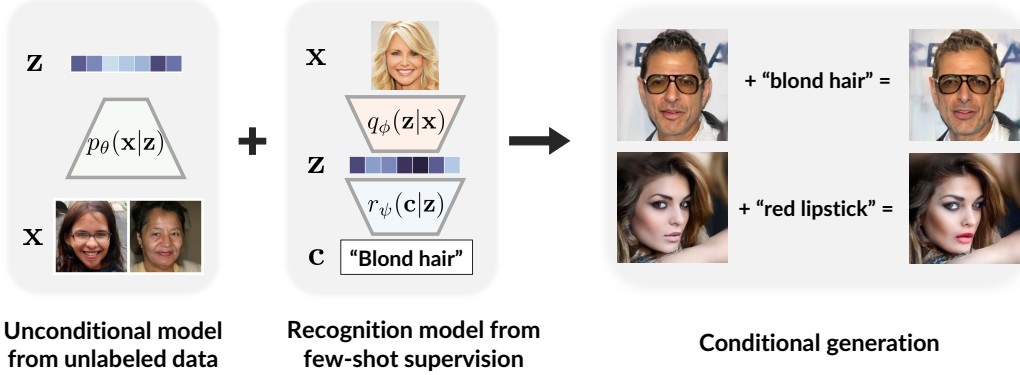

Figure 1: Few-shot conditional generation with the unconditional D2C model (left). With a recognition model over the latent space (middle), D2C can generate samples for novel conditions, such as image manipulation (right). These conditions can be defined with very few labels.

In this paper, we propose Diffusion-Decoding models with Contrastive representations (D2C), a special VAE that is suitable for conditional few-shot generation. D2C uses contrastive self-supervised learning methods to obtain a latent space that inherits the transferrability and few-shot capabilities of self-supervised representations. Unlike other VAEs, D2C learns a diffusion model over the latent representations. This latent diffusion model ensures that D2C uses the same latent distribution for both training and generation. We provide a formal argument to explain why this approach leads to better sample quality than existing hierarchical VAEs. We further discuss how to apply D2C to few-shot conditional generation where the conditions are defined through labeled examples or manipulation constraints. Our approach combines a discriminative model providing conditioning signal and generative diffusion model over the latent space, and is computationally more efficient than methods that act directly over the image space (Figure 1).

We evaluate and compare D2C with several state-of-the-art generative models over 6 datasets. On unconditional generation, D2C outperforms state-of-the-art VAEs and is competitive with diffusion models under similar computational budgets. On conditional generation with 100 labeled examples, D2C significantly outperforms state-of-the-art VAE [91] and diffusion models [84]. D2C can also learn to perform certain image manipulation tasks from as few as 100 labeled examples. Notably, for manipulating images, D2C is two orders of magnitude faster than StyleGAN2 [106] and preferred by $50\% - 60\%$ of human evaluations, which to our best knowledge is the first for any VAE model.

## 2 Background

**Latent variable generative models**  A latent variable generative model (LVGM) is posed as a conditional distribution $p_\theta : \mathcal{Z} \to \mathcal{P}(\mathcal{X})$ from a latent variable $\mathbf{z}$ to a generated sample $\mathbf{x}$, parametrized by $\theta$. To acquire new samples, LVGMs draw random latent variables $\mathbf{z}$ from some distribution $p(\mathbf{z})$ and map them to image samples through $\mathbf{x} \sim p_\theta(\mathbf{x}|\mathbf{z})$. Most LVGMs are built on top of four paradigms: variational autoencoders (VAEs, [52, 77]), Normalizing Flows (NFs, [28, 29]), Generative Adversarial Networks (GANs, [34]), and diffusion / score-based generative models [42, 85].

Particularly, VAEs use an inference model from $\mathbf{x}$ to $\mathbf{z}$ for training. Denoting the inference distribution from $\mathbf{x}$ to $\mathbf{z}$ as $q_\phi(\mathbf{z}|\mathbf{x})$, the generative distribution from $\mathbf{z}$ to $\mathbf{x}$ as $p_\theta(\mathbf{x}|\mathbf{z})$, VAEs are trained by minimizing the following upper bound of negative log-likelihood:

$$L_{\text{VAE}} = \mathbb{E}_{\mathbf{x} \sim p_{\text{data}}}[\mathbb{E}_{\mathbf{z} \sim q_\phi(\mathbf{z}|\mathbf{x})}[-\log p(\mathbf{x}|\mathbf{z})] + D_{\text{KL}}(q_\phi(\mathbf{z}|\mathbf{x})\|p(\mathbf{z}))] \qquad (1)$$

where $p_{\text{data}}$ is the data distribution and $D_{\text{KL}}$ is the KL-divergence.

**Diffusion models**   Diffusion models [81, 42, 84] produce samples by reversing a Gaussian diffusion process. We use the index $\alpha \in [0, 1]$ to denote the particular noise level of an noisy observation $\mathbf{x}^{(\alpha)} = \sqrt{\alpha}\mathbf{x} + \sqrt{1 - \alpha}\epsilon$, where $\mathbf{x}$ is the clean observation and $\epsilon \sim \mathcal{N}(0, \boldsymbol{I})$ is a standard Gaussian distribution; as $\alpha \to 0$, the distribution of $\mathbf{x}^{(\alpha)}$ converges to $\mathcal{N}(0, \boldsymbol{I})$. Diffusion models are typically parametrized as reverse noise models $\epsilon_\theta(\mathbf{x}^{(\alpha)}, \alpha)$ that predict the noise component of $\mathbf{x}^{(\alpha)}$ given a noise level $\alpha$, and trained to minimize $\|\epsilon_\theta(\mathbf{x}^{(\alpha)}, \alpha) - \epsilon\|_2^2$, the mean squared error loss between the true noise and predicted noise. Given any non-increasing series $\{\alpha_i\}_{i=0}^T$ between 0 and 1, the diffusion objective for a clean sample from the data $\mathbf{x}$ is:

$$\ell_{\text{diff}}(\mathbf{x}; w, \theta) := \sum_{i=1}^T w(\alpha_i)\mathbb{E}_{\epsilon \sim \mathcal{N}(0, \boldsymbol{I})}[\|\epsilon - \epsilon_\theta(\mathbf{x}^{(\alpha_i)}, \alpha_i)\|_2^2], \quad \mathbf{x}^{(\alpha_i)} := \sqrt{\alpha_i}\mathbf{x} + \sqrt{1 - \alpha_i}\epsilon \quad (2)$$

where $w : \{\alpha_i\}_{i=1}^T \to \mathbb{R}_+$ controls the loss weights for each $\alpha$. When $w(\alpha) = 1$ for all $\alpha$, we recover the denoising score matching objective for training score-based generative models [85].

Given an initial sample $\mathbf{x}_0 \sim \mathcal{N}(0, \boldsymbol{I})$, diffusion models acquires clean samples (*i.e.*, samples of $\mathbf{x}_1$) through a gradual denoising process, where samples with reducing noise levels $\alpha$ are produced (*e.g.*, $\mathbf{x}_0 \to \mathbf{x}_{0.3} \to \mathbf{x}_{0.7} \to \mathbf{x}_1$). In particular, Denoising Diffusion Implicit Models (DDIMs, [84]) uses an Euler discretization of some neural ODE [13] to produce samples (Figure 2, left).

We provide a more detailed description for training diffusion models in Appendix A.1 and sampling from DDIM in Appendix A.2. For conciseness, we use the notation $p^{(\alpha)}(\mathbf{x}^{(\alpha)})$ to denote the marginal distribution of $\mathbf{x}^{(\alpha)}$ under the diffusion model, and $p^{(\alpha_1, \alpha_2)}(\mathbf{x}^{(\alpha_2)} \mid \mathbf{x}^{(\alpha_1)})$ to denote the diffusion sampling process from $\mathbf{x}^{(\alpha_1)}$ to $\mathbf{x}^{(\alpha_2)}$ (assuming $\alpha_1 < \alpha_2$).

**Self-supervised learning of representations**   In self-supervised learning (SSL), representations are learned by completing certain pretext tasks that do not require extra manual labeling [68, 25]; these representations can then be applied to other downstream tasks, often in few-shot or zero-shot scenarios. In particular, contrastive representation learning encourages representations to be closer between "positive" pairs and further between "negative" pairs; contrastive predictive coding (CPC, [95]), based on multi-class classification, have been commonly used in state-of-the-art methods [37, 16, 18, 14, 82]. Other non-contrastive methods exist, such as BYOL [35] and SimSiam [17], but they usually require additional care to prevent the representation network from collapsing.

## 3   Problem Statement

**Few-shot conditional generation**   Our goal is to learn an unconditional generative model $p_\theta(\mathbf{x})$ such that it is suitable for conditional generation. Let $\mathcal{C}(\mathbf{x}, \mathbf{c}, f)$ describe an event that "$f(\mathbf{x}) = \mathbf{c}$", where $\mathbf{c}$ is a property value and $f(\mathbf{x})$ is a property function that is unknown at training. In conditional generation, our goal is to sample $\mathbf{x}$ such that the event $\mathcal{C}(\mathbf{x}, \mathbf{c}, f)$ occurs for a chosen $\mathbf{c}$. If we have access to some "ground-truth" model that gives us $p(\mathcal{C}|\mathbf{x}) := p(f(\mathbf{x}) = \mathbf{c}|\mathbf{x})$, then the conditional model can be derived from Bayes' rule: $p_\theta(\mathbf{x}|\mathcal{C}) \propto p(\mathcal{C}|\mathbf{x})p_\theta(\mathbf{x})$. These properties $\mathbf{c}$ include (but are not limited to[2]) labels [61], text descriptions [60, 76], noisy or partial observations [11, 5, 47, 24], and manipulation constraints [69]. In many cases, we do not have direct access to the true $f(\mathbf{x})$, so we need to learn an accurate model from labeled data [6] (*e.g.*, $(\mathbf{c}, \mathbf{x})$ pairs).

**Desiderata**   Many existing methods are optimized for some known condition (*e.g.*, labels in conditional GANs [8]) or assume abundant pairs between images and conditions that can be used for pretraining (*e.g.*, DALL-E [74] and CLIP [73] over image-text pairs). Neither is the case in this paper, as we do not expect to train over paired data.

While high-quality latent representations are not essential to unconditional image generation (*e.g.*, autoregressive [94], energy-based [31], and some diffusion models [42]), they can be beneficial when we wish to specify certain conditions with limited supervision signals, similar to how SSL representations can reduce labeling efforts in downstream tasks. A compelling use case is detecting and removing biases in datasets via image manipulation, where we should not only address well-known biases a-priori but also address other hard-to-anticipate biases, adapting to societal needs [65].

---

[2]When $\mathcal{C}$ refers to an event that is always true, we recover unconditioned generation.

Table 1: A comparison of several common paradigms for generative modeling. [Explicit $\mathbf{x} \to \mathbf{z}$]: the mapping from $\mathbf{x}$ to $\mathbf{z}$ is directly trainable, which enables SSL; [No prior hole]: latent distributions used for generation and training are identical (Sec. 4.2), which improves generation; [Non-adversarial]: training procedure does not involve adversarial optimization, which improves training stability.

| Model family | Explicit $\mathbf{x} \to \mathbf{z}$ (Enables SSL) | No prior hole (Better generation) | Non-Adversarial (Stable training) |
|---|---|---|---|
| VAE [52, 77], NF [28] | ✔ | ✗ | ✔ |
| GAN [34] | ✗ | ✔ | ✗ |
| BiGAN [30, 32] | ✔ | ✔ | ✗ |
| DDIM [84] | ✗ | ✔ | ✔ |
| **D2C** | ✔ | ✔ | ✔ |

Therefore, a desirable generative model should not only have high sample quality but also contain informative latent representations. While VAEs are ideal for learning rich latent representations due to being able to incorporate SSL within the encoder, they generally do not achieve the same level of sample quality as GANs and diffusion models.

## 4 Diffusion-Decoding Generative Models with Contrastive Learning

In this paper, we present Diffusion-Decoding generative models with Contrastive Learning (D2C), an extension to VAEs with high-quality samples and high-quality latent representations, and are thus well suited to few-shot conditional generation. Moreover, unlike GAN-based methods, D2C does not involve unstable adversarial training (Table 1).

As its name suggests, the generative model for D2C has two components – *diffusion* and *decoding*; the *diffusion* component operates over the latent space and the *decoding* component maps from latent representations to images. Let us use the $\alpha$ index notation for diffusion random variables: $\mathbf{z}^{(0)} \sim p^{(0)}(\mathbf{z}^{(0)}) := \mathcal{N}(0, \boldsymbol{I})$ is the "noisy" latent variable with $\alpha = 0$, and $\mathbf{z}^{(1)}$ is the "clean" latent variable with $\alpha = 1$. The generative process of D2C, which we denote $p_\theta(\mathbf{x}|\mathbf{z}^{(0)})$, is then defined as:

$$\mathbf{z}^{(0)} \sim p^{(0)}(\mathbf{z}^{(0)}), \quad \mathbf{z}^{(1)} \sim \underbrace{p_\theta^{(0,1)}(\mathbf{z}^{(1)}|\mathbf{z}^{(0)})}_{\text{diffusion}}, \quad \mathbf{x} \sim \underbrace{p_\theta(\mathbf{x}|\mathbf{z}^{(1)})}_{\text{decoding}}, \tag{3}$$

where $p^{(0)}(\mathbf{z}^{(0)}) = \mathcal{N}(0, \boldsymbol{I})$ is the prior distribution for the diffusion model, $p_\theta^{(0,1)}(\mathbf{z}^{(1)}|\mathbf{z}^{(0)})$ is the diffusion process from $\mathbf{z}^{(0)}$ to $\mathbf{z}^{(1)}$, and $p_\theta(\mathbf{x}|\mathbf{z}^{(1)})$ is the decoder from $\mathbf{z}^{(1)}$ to $\mathbf{x}$. Intuitively, D2C models produce samples by drawing $\mathbf{z}^{(1)}$ from a diffusion process and then decoding $\mathbf{x}$ from $\mathbf{z}^{(1)}$.

In order to train a D2C model, we use an inference model $q_\phi(\mathbf{z}^{(1)}|\mathbf{x})$ that predicts proper $\mathbf{z}^{(1)}$ latent variables from $\mathbf{x}$; this can directly incorporate SSL methods [99], leading to the following objective:

$$L_{\text{D2C}}(\theta, \phi; w) := L_{\text{D2}}(\theta, \phi; w) + \lambda L_{\text{C}}(q_\phi), \tag{4}$$

$$L_{\text{D2}}(\theta, \phi; w) := \mathbb{E}_{\mathbf{x} \sim p_{\text{data}}, \mathbf{z}^{(1)} \sim q_\phi(\mathbf{z}^{(1)}|\mathbf{x})}[-\log p_\theta(\mathbf{x}|\mathbf{z}^{(1)}) + \ell_{\text{diff}}(\mathbf{z}^{(1)}; w, \theta)], \tag{5}$$

where $\ell_{\text{diff}}$ is defined as in Eq.(2), $L_C(q_\phi)$ denotes any contrastive predictive coding objective [95] with rich data augmentations [37, 16, 18, 14, 82] (details in Appendix A.3) and $\lambda > 0$ is a weight hyperparameter. We illustrate D2C in Figure 2, and its training procedure in Appendix A.4.

### 4.1 Relationship to maximum likelihood

The D2 objective ($L_{\text{D2}}$) appears similar to the original VAE objective ($L_{\text{VAE}}$). Here, we make an informal statement that the D2 objective function is deeply connected to the variational lower bound of log-likelihood; we present the full statement and proof in Appendix B.1.

**Theorem 1.** *(informal) For any valid $\{\alpha_i\}_{i=0}^T$, there exists some weights $\hat{w} : \{\alpha_i\}_{i=1}^T \to \mathbb{R}_+$ for the diffusion objective such that $-L_{\text{D2}}$ is a variational lower bound to the log-likelihood, i.e.,*

$$-L_{\text{D2}}(\theta, \phi; \hat{w}) \leq \mathbb{E}_{p_{\text{data}}}[\log p_\theta(\mathbf{x})], \tag{6}$$

*where $p_\theta(\mathbf{x}) := \mathbb{E}_{\mathbf{z}_0 \sim p^{(0)}(\mathbf{z}^{(0)})}[p_\theta(\mathbf{x}|\mathbf{z}^{(0)})]$ is the marginal probability of $\mathbf{x}$ under the D2C model.*

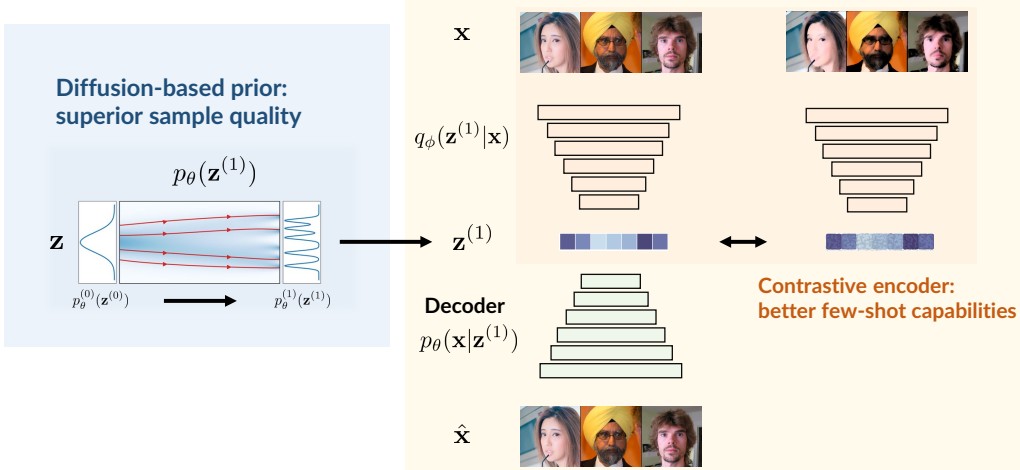

Figure 2: Illustration of components of a D2C model. On top of the encoding and decoding between $\mathbf{x}$ and $\mathbf{z}^{(1)}$, we use a diffusion model to generate $\mathbf{z}^{(1)}$ from a Gaussian $\mathbf{z}^{(0)}$. The red lines in the diffusion prior describe several smooth ODE trajectories from $\alpha = 0$ to $\alpha = 1$.

*Proof.* (sketch) The diffusion term $\ell_{\mathrm{diff}}$ upper bounds the KL divergence between $q_\phi(\mathbf{z}_1|\mathbf{x})$ and $p_\theta^{(1)}(\mathbf{z}^{(1)})$ for suitable weights [42, 84], which recovers a VAE objective. $\qquad\square$

Moreover, since some contrastive losses are lower bounds to mutual information [82], D2C can be treated as an implementation of the InfoVAE approach [103], where the latent variables are encouraged to have higher mutual information with the observations. Since the evidence lower bound (ELBO) for VAEs do not necessarily promote informative latent representations [15], D2C comes with the benefit of being able to learn more informative latents despite not exactly optimizing an ELBO.

### 4.2   D2C models address latent posterior mismatch in VAEs

While D2C is a special case of VAE, we argue that D2C is non-trivial in the sense that it addresses a long-standing problem in VAE methods [90, 87], namely the mismatch between the prior distribution $p_\theta(\mathbf{z})$ and the aggregate (approximate) posterior distribution $q_\phi(\mathbf{z}) := \mathbb{E}_{p_{\mathrm{data}}(\mathbf{x})}[q_\phi(\mathbf{z}|\mathbf{x})]$. A mismatch could create "holes" [79, 43, 3] in the prior that the aggregate posterior fails to cover during training, resulting in worse sample quality, as many latent variables used during generation are likely to never have been trained on. We formalize this notion in the following definition.

**Definition 1** (Prior hole). *Let $p(\mathbf{z}), q(\mathbf{z})$ be two distributions with $\mathrm{supp}(q) \subseteq \mathrm{supp}(p)$. We say that $q$ has an $(\epsilon, \delta)$-**prior hole** with respect to (the prior) $p$ for $\epsilon, \delta \in (0, 1)$, $\delta > \epsilon$, if there exists a set $S \in \mathrm{supp}(P)$, such that $\int_S p(\mathbf{z})\mathrm{d}\mathbf{z} \geq \delta$ and $\int_S q(\mathbf{z})\mathrm{d}\mathbf{z} \leq \epsilon$.*

Intuitively, if $q_\phi(\mathbf{z})$ has a prior hole with large $\delta$ and small $\epsilon$ (*e.g.*, inversely proportional to the number of training samples), then it is very likely that latent variables within the hole are never seen during training (small $\epsilon$), yet frequently used to produce samples (large $\delta$). Most existing methods address this problem by optimizing certain statistical divergences between $q_\phi(\mathbf{z})$ and $p_\theta(\mathbf{z})$, such as the KL divergence or Wasserstein distance [88]. However, we argue in the following statement that prior holes might not be eliminated even if we optimize certain divergence values to be reasonably low, especially when $q_\phi(\mathbf{z})$ is very flexible. We present the formal statement and proof in Appendix B.2.

**Theorem 2.** *(informal) Let $p_\theta(\mathbf{z}) = \mathcal{N}(0, 1)$. For any $\epsilon > 0$, there exists a $q_\phi(\mathbf{z})$ with an $(\epsilon, 0.49)$-prior hole, such that $D_{\mathrm{KL}}(q_\phi \| p_\theta) \leq \log 2$[3] and $W_2(q_\phi, p_\theta) < \gamma$ for any $\gamma > 0$, where $W_2$ is the 2-Wasserstein distance.* 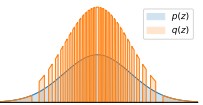

---

[3]This is reasonably low for realistic VAE models (NVAE [91] reports a KL divergence of around 2810 nats).

*Proof.* (sketch) We construct a $q_\phi$ that satisfies these properties (top-right figure). First, we truncate the Gaussian and divide them into regions with same probability mass; then we support $q_\phi$ over half of these regions (so $\delta > 0.49$); finally, we show that the divergences are small enough. □

In contrast to addressing prior holes by optimization, diffusion models eliminate prior holes by construction, since the diffusion process from $\mathbf{z}^{(1)}$ to $\mathbf{z}^{(0)}$ is constructed such that the distribution of $\mathbf{z}^{(\alpha)}$ always converges to a standard Gaussian as $\alpha \to 0$. As a result, the distribution of latent variables used during training is arbitrarily close to that used in generation[4], which is also the case in GANs. Therefore, our argument provides an explanation as to why we observe better sample quality results from GANs and diffusion models than VAEs and NFs.

## 5  Few-shot Conditional Generation with D2C

In this section, we discuss how D2C can be used to learn to perform conditional generation from few-shot supervision. We note that D2C is only trained on images and not with any other data modalities (*e.g.*, image-text pairs [74]) or supervision techniques (*e.g.*, meta-learning [22, 6]).

**Algorithm**  We describe the general algorithm for conditional generation from a few images in Algorithm 1, and detailed implementations in Appendix C. With a model over the latent space (denoted as $r_\psi(\mathbf{c}|\mathbf{z}^{(1)})$), we draw conditional latents from an unnormalized distribution with the diffusion prior (line 4). This can be implemented in many ways such as rejection sampling or Langevin dynamics [66, 86, 27].

---

**Algorithm 1** Conditional generation with D2C

1: **Input** $n$ examples $\{(\mathbf{x}_i, \mathbf{c}_i)\}_{i=1}^n$, property $\mathbf{c}$.
2: Acquire latents $\mathbf{z}_i^{(1)} \sim q_\phi(\mathbf{z}^{(1)}|\mathbf{x}_i)$ for $i \in [n]$;
3: Train model $r_\psi(\mathbf{c}|\mathbf{z}^{(1)})$ over $\{(\mathbf{z}_i^{(1)}, \mathbf{c}_i)\}_{i=1}^n$
4: Sample latents with $\hat{\mathbf{z}}^{(1)} \sim r_\psi(\mathbf{c}|\mathbf{z}^{(1)}) \cdot p_\theta^{(1)}(\mathbf{z}^{(1)})$ (unnormalized);
5: Decode $\hat{\mathbf{x}} \sim p_\theta(\mathbf{x}|\hat{\mathbf{z}}^{(1)})$.
6: **Output** $\hat{\mathbf{x}}$.

---

**Conditions from labeled examples**  Given a few labeled examples, we wish to produce diverse samples with a certain label. For labeled examples we can directly train a classifier over the latent space, which we denote as $r_\psi(\mathbf{c}|\mathbf{z}^{(1)})$ with $\mathbf{c}$ being the class label and $\mathbf{z}^{(1)}$ being the latent representation of $\mathbf{x}$ from $q_\phi(\mathbf{z}^{(1)}|\mathbf{x})$. If these examples do not have labels (*i.e.*, we merely want to generate new samples similar to given ones), we can train a positive-unlabeled (PU) classifier [33] where we assign "positive" to the new examples and "unlabeled" to training data. Then we use the classifier with the diffusion model $p_\theta(\mathbf{z}^{(1)}|\mathbf{z}^{(0)})$ to produce suitable values of $\mathbf{z}^{(1)}$, such as by rejecting samples from the diffusion model that has a small $r_\psi(\mathbf{c}|\mathbf{z}^{(1)})$.

**Conditions from manipulation constraints**  Given a few labeled examples, here we wish to learn how to manipulate images. Specifically, we condition over the event that "$\mathbf{x}$ has label $\mathbf{c}$ but is similar to image $\bar{\mathbf{x}}$". Here $r_\psi(\mathbf{c}|\mathbf{z}^{(1)})$ is the unnormalized product between the classifier conditional probability and closeness to the latent $\bar{\mathbf{z}}^{(1)}$ of $\bar{\mathbf{x}}$ (*e.g.*, measured with RBF kernel). We implement line 4 of Alg. 1 with a Lanvegin-like procedure where we take a gradient step with respect to the classifier probability and then correct this gradient step with the diffusion model. Unlike many GAN-based methods [12, 72, 97, 45, 98], D2C does not need to optimize an inversion procedure at evaluation time, and thus the latent value is much faster to compute; D2C is also better at retaining fine-grained features of the original image due to the reconstruction loss.

## 6  Related Work

**Latent variable generative models**  Most deep generative models explicitly define a latent representation, except for some energy-based models [41, 31] and autoregressive models [94, 93, 10]. Unlike VAEs and NFs, GANs do not explicitly define an inference model and instead optimize a two-player game. In terms of sample quality, GANs currently achieve superior performance over VAEs and NFs, but they can be difficult to invert even with additional optimization [48, 100, 7]. This can be partially addressed by training reconstruction-based losses with GANs [54, 55]. Moreover,

---

[4]We expand this argument in Appendix B.2.

the GAN training procedure can be unstable [9, 8, 63], lack a informative objective for measuring progress [4], and struggle with discrete data [102]. Diffusion models [27] achieves high sample quality without adversarial training, but its latent dimension must be equal to the image dimension. A concurrent work [92] applied score-based generative modeling on the latent space of an VAE, and achieved similar improvements compared to regular VAEs.

**Addressing posterior mismatch in VAEs** Most methods address this mismatch problem by improving inference models [64, 51, 89], prior models [90, 3, 87], or objective functions [103, 104, 105, 1, 59]; all these approaches optimize the posterior model to be close to the prior. In Section 4.2, we explain why these approaches do not necessarily remove large "prior holes", so their sample qualities remain relatively poor even after many layers [91, 19]. Other methods adopt a "two-stage" approach [23], which fits a generative model over the latent space of autoencoders [96, 75, 26, 74]. [62] have applied diffusion models directly on latent spaces learned by a music VAE.

**Conditional generation with unconditional models** To perform conditional generation over an unconditional LVGM, most methods assume access to a discriminative model (*e.g.*, a classifier); the latent space of the generator is then modified to change the outputs of the discriminative model. The disciminative model can operate on either the image space [66, 70, 27] or the latent space [80, 98]. For image space discriminative models, plug-and-play generative networks [66] control the attributes of generated images via Langevin dynamics [78]; these ideas are also explored in diffusion models [86]. Image manipulation methods are based on GANs often operate with latent space discriminators [80, 98]. However, these methods have some trouble manipulating real images because of imperfect reconstruction [107, 7]. This is not a problem in D2C since a reconstruction objective is optimized.

## 7 Experiments

We examine the conditional and unconditional generation qualities of D2C over CIFAR-10 [53], CIFAR-100 [53], fMoW [21], CelebA-64 [58], CelebA-HQ-256 [48], and FFHQ-256 [49]. Our D2C implementation is based on the state-of-the-art NVAE [91] autoencoder structure, the U-Net diffusion model [42], and the MoCo-v2 contrastive representation learning method [16]. We also consider the D2 objective where we do not applying the contrastive loss in the NVAE autoencoder; we also tried adding contrastive loss directly to the VAE objective, but we were unable to achieve satisfactory generation results (reconstruction MSE remains high). This is possibly due to the many regularizations needed for NVAE to work well, which could conflict with contrastive learning[5].

We keep the diffusion series hyperparameter $\{\alpha_i\}_{i=1}^T$ identical to ensure a fair comparison with different diffusion models. For the contrastive weight $\lambda$ in Equation (4), we consider the value of $\lambda = 10^{-4}$ based on the relative scale between the $L_C$ and $L_{D2}$; we find that the results are relatively insensitive to $\lambda$. We use 100 diffusion steps for DDIM and D2C unless mentioned otherwise, as running with longer steps is not computationally economical despite tiny gains in FID [84]. We include additional training details, such as architectures, optimizers and learning rates in Appendix C.

Table 2: Quality of representations and generations with LVGMs.

| Model | CIFAR-10 | | | CIFAR-100 | | | fMoW | | |
|---|---|---|---|---|---|---|---|---|---|
| | FID ↓ | MSE ↓ | Acc ↑ | FID ↓ | MSE ↓ | Acc ↑ | FID ↓ | MSE ↓ | Acc ↑ |
| NVAE [91] | 36.4 | 0.25 | 18.8 | 42.5 | 0.53 | 4.1 | 82.25 | **0.30** | 27.7 |
| DDIM [84] | **4.16** | 2.5 | 22.5 | **10.16** | 3.2 | 2.2 | **37.74** | 3.0 | 23.5 |
| D2 (Ours) | 15.1 | **0.24** | 40.6 | 19.85 | 0.48 | 17.89 | - | - | - |
| D2C (Ours) | 10.15 | 0.76 | **76.02** | 14.62 | **0.44** | **42.75** | 44.7 | 2.33 | **66.9** |

### 7.1 Unconditional generation

For unconditional generation, we measure the sample quality of images using the Frechet Inception Distance (FID, [40]) with 50,000 images. In particular, we extensively evaluate NVAE [91] and

---

[5]See `https://github.com/NVlabs/NVAE#known-issues` for a detailed description.

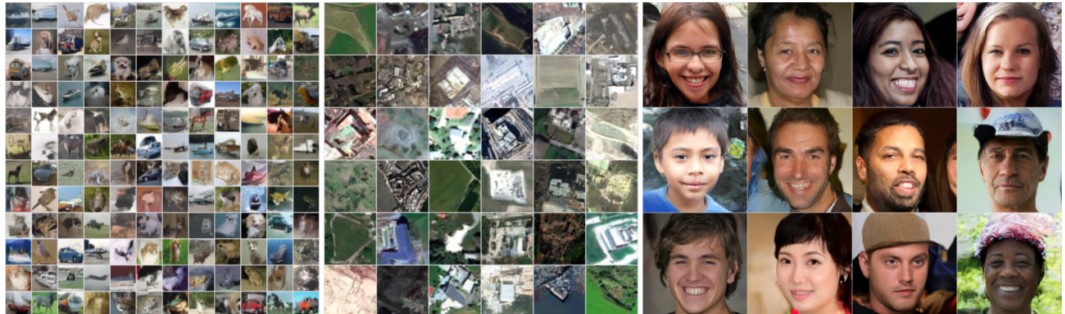

Figure 3: Generated samples on CIFAR-10 (left), fMoW (mid), and FFHQ $256 \times 256$ (right).

DDIM [84], a competitive VAE model and a competitive diffusion model as baselines because we can directly obtain features from them without additional optimization steps[6]. For them, we report mean-squared reconstruction error (MSE, summed over all pixels, pixels normalized to $[0, 1]$) and linear classification accuracy (Acc., measured in percentage) over $\mathbf{z}_1$ features for the test set.

We report sample quality results[7] in Tables 2, and 3. For FID, we outperform NVAE in all datasets and outperform DDIM on CelebA-64 and CelebA-HQ-256, which suggests our results are competitive with state-of-the-art non-adversarial generative models. In Table 2, we additionally compare NVAE, DDIM and D2C in terms of reconstruction and linear classification accuracy. As all three methods contain reconstruction losses, the MSE values are low and comparable. However, D2C enjoys much better linear classification accuracy than the other two thanks to the contrastive SSL component. We further note that training the same contrastive SSL method without $L_{\mathrm{D2}}$ achieves slightly higher 78.3% accuracy on CIFAR-10. We tried improving this via ResNet [38] encoders, but this significantly increased reconstruction error, possibly due to loss of information in average pooling layers.

Table 3: FID scores over different faces dataset with LVGMs.

| Model | CelebA-64 | CelebA-HQ-256 | FFHQ-256 |
|---|---|---|---|
| NVAE [91] | 13.48 | 40.26 | 26.02 |
| DDIM [84] | 6.53 | 25.6 | - |
| D2C (Ours) | **5.7** | **18.74** | **13.04** |

### 7.2 Few-shot conditional generation from examples

We demonstrate the advantage of D2C representations by performing few-shot conditional generation over labels. We consider two types of labeled examples: one has binary labels for which we train a binary classifier; the other is positive-only labeled (*e.g.*, images of female class) for which we train a PU classifier. Our goal here is to generate a diverse group of images with a certain label. We evaluate and compare three models: D2C, NVAE and DDIM. We train a classifier $r_\psi(\mathbf{c}|\mathbf{z})$ over the latent space of these models; we also train a image space classifier and use it with DDIM (denoted as DDIM-I). We run Algorithm 1 for these models, where line 4 is implemented via rejection sampling. As our goal is to compare different models, we leave more sophisticated methods [27] as future work.

First, we consider performing 8 conditional generation tasks over CelebA-64 with 2 binary classifiers (trained over 100 samples, 50 for each class) and 4 PU classifiers (trained over 100 positively labeled and 10k unlabeled samples). We also report a "naive" approach where we use all the training images (regardless of labels) and compute its FID with the corresponding subset of images (*e.g.*, all images versus blond images). In Table 4, we report the FID score between generated images (5k samples) and real images of the corresponding label.

Next, we perform a similar experiment on CIFAR-10, but with 50 labels for each of the 10 classes. For each label, we evaluate the FID between the real images and generated images under that label

---

[6]For DDIM, the latent representations $\mathbf{x}^{(0)}$ are obtained by reversing the neural ODE process.

[7]Due to space limits, we place additional CIFAR-10 results in Appendix D.

and take the average FID over the 10 labels. The average FID for D2C, DDIM, and NVAE are 32.57, 82.78, and 95.45 respectively (note that FIDs are expected to be higher because it is evaluated with 5k real samples per class instead of the commonly used 50k). This shows that D2C performs significantly better than DDIM and NVAE on the few-shot conditional generation task for both datasets, illustrating the advantage of contrastive representations for few-shot conditional generation. We demonstrate some figures in the appendix.

Table 4: FID scores for few-shot conditional generation with various types of labeled examples. Naive performs very well for non-blond due to class percentages.

| Method | Classes (% in train set) | D2C | DDIM | NVAE | DDIM-I | Naive |
|---|---|---|---|---|---|---|
| Binary | Male (42%) | **13.44** | 38.38 | 41.07 | 29.03 | 26.34 |
| | Female (58%) | **9.51** | 19.25 | 16.57 | 15.17 | 18.72 |
| | Blond (15%) | **17.61** | 31.39 | 31.24 | 29.09 | 27.51 |
| | Non-Blond (85%) | **8.94** | 9.67 | 16.73 | 19.76 | 3.77 |
| PU | Male (42%) | **16.39** | 37.03 | 42.78 | 19.60 | 26.34 |
| | Female (58%) | **12.21** | 15.42 | 18.36 | 14.96 | 18.72 |
| | Blond (15%) | **10.09** | 30.20 | 31.06 | 76.52 | 27.51 |
| | Non-Blond (85%) | **9.09** | 9.70 | 17.98 | 9.90 | 3.77 |

**Fair generative modeling with D2C** We illustrate some conditional generation results with D2C (from 100 "blond" labels) in Figure 4 (left), which exhibits a biased gender ratio (3 examples out of 36). This kind of dataset bias in CelebA is well-documented in previous work [46]; among all the images with the "blond" attribute, only 5.6 are labeled as "male". In fact, we observe that the D2C samples have a similar gender bias as the original dataset since D2C has learned the data distribution quite well. To mitigate the presence of dataset bias in D2C's samples, we leverage a conditional generation approach. This is in contrast to previous methods that try to learn a fair generative model via a carefully designed representative dataset [20]. We give a concrete instantiation of our method in the case of gender bias that can be generalized to all attributes. Suppose our goal is to achieve a balanced gender ratio among all the generated images with "blond" attribute; we can do so by generating the same number of samples conditioned on the (1) "blond, male" attributes and the (2) "blond, female" attributes. Figure 4 (right) demonstrates that D2C can be used for fair generative modeling by simply specifying our desired constraints with a handful of labels.

Blond

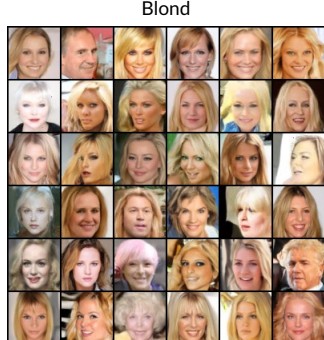 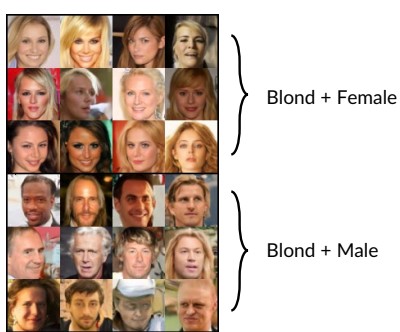

Blond + Female

Blond + Male

Figure 4: (Left) While D2C generations are faithful to the CelebA distributions, it also learns from the biases within the dataset. (Right) Conditional generation can make D2C a fairer generative model.

### 7.3 Few-shot conditional generation from manipulation constraints

Finally, we consider image manipulation where we use binary classifiers that are learned over 50 labeled instances for each class. To manipulate an image, we first acquire its latent code, move its latent code towards the decision boundary of the classifier and then use the diffusion prior to make the

| Original | D2C | StyleGAN2 | NVAE | DDIM |
|----------|-----|-----------|------|------|

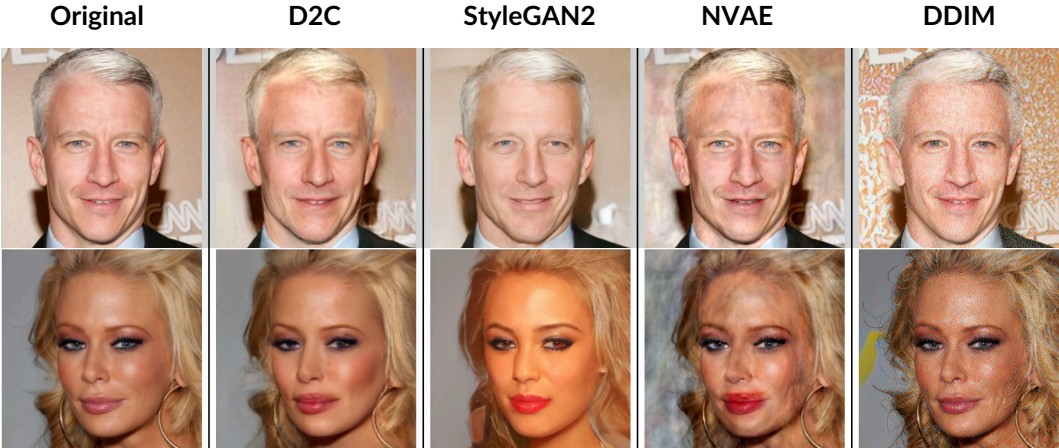

Figure 5: Image manipulation results for *blond* (top) and *red lipstick* (bottom). D2C is better than StyleGAN2 at preserving details of the original image, such as eyes, earrings, and background.

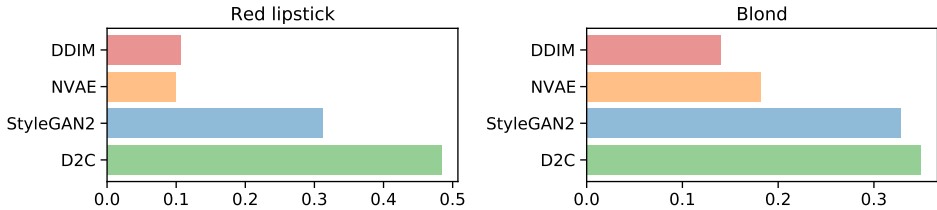

Figure 6: AMT evaluation over image manipulations. $x$-axis shows the percentage that the evaluator selects the image generated from the corresponding model out of 4 images from each model.

sample more realistic. We perform Amazon Mechanical Turk (AMT) evaluations over two attributes in the CelebA-256 dataset, *blond* and *red lipstick*, over D2C, DDIM, NVAE and StyleGAN2 [50] (see Figure 5). The evaluation is double-blinded: neither we nor the evaluators know the correspondence between generated image and underlying model during the study. We include more details (algorithm, setup and human evaluation) in Appendix C and additional qualitative results (such as *beard* and *gender* attributes) in Appendix D.

In Figure 8, we show the percentage of manipulations preferred by AMT evaluators for each model; D2C slightly outperforms StyleGAN2 for *blond* and significantly outperforms StyleGAN2 for *red lipstick*. When we compare D2C with only StyleGAN2, D2C is preferred over $51.5\%$ for *blond* and $60.8\%$ for *red lipstick*. An additional advantage of D2C is that the manipulation is much faster than StyleGAN2, since the latter requires additional optimization over the latent space to improve reconstruction [106]. On the same Nvidia 1080Ti GPU, it takes 0.013 seconds to obtain the latent code in D2C, while the same takes 8 seconds [106] for StyleGAN2 ($615\times$ slower). As decoding is very fast for both models, D2C generations are around two orders of magnitude faster to produce.

## 8    Discussions and Limitations

We introduced D2C, a VAE-based generative model with a latent space suitable for few-shot conditional generation. To our best knowledge, our model is the first unconditional VAE to demonstrate superior image manipulation performance than StyleGAN2, which is surprising given our use of a regular NVAE architecture. We believe that with better architectures, such as designs from Style-GAN2 or Transformers [44], D2C can achieve even better performance. It is also interesting to formally investigate the integration between D2C and other types of conditions on the latent space, as well as training D2C in conjunction with other domains and data modalities, such as text [74], in a fashion that is similar to semi-supervised learning. Nevertheless, we note that our model have to be used properly in order to mitigate potential negative societal impacts, such as deep fakes.

## Acknowledgements

We thank the anonymous reviewers and Kristy Choi for insightful discussions and feedback.

**Funding transparency statement.** This research was supported by NSF (#1651565, #1522054, #1733686), ONR (N00014-19-1-2145), AFOSR (FA9550-19-1-0024), ARO (W911NF-21-1-0125), Sloan Fellowship, Amazon AWS, Stanford Institute for Human-Centered Artificial Intelligence (HAI), and Google Cloud.

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
