# A  Additional Details for D2C

## A.1  Training diffusion models

We use the notations in [84] to denote the $\alpha$ values and consider the forward diffusion model in [42]; a non-Markovian version that motivates other sampling procedures can be found in [84], but the training procedure is largely identical. We refer to the reader to these two papers for more details.

First, we define the following diffusion forward process for a series $\{\alpha_t\}_{t=0}^T$:

$$q(\mathbf{x}^{(\alpha_{1:T})}|\mathbf{x}^{(\alpha_0)}) := \prod_{t=1}^T q(\mathbf{x}^{(\alpha_t)}|\mathbf{x}^{(\alpha_{t-1})}), \tag{7}$$

$$q(\mathbf{x}^{(\alpha_t)}|\mathbf{x}^{(\alpha_{t-1})}) := \mathcal{N}\left(\sqrt{\frac{\alpha_t}{\alpha_{t-1}}}\mathbf{x}^{(\alpha_{t-1})}, \left(1 - \frac{\alpha_t}{\alpha_{t-1}}\right)\mathbf{I}\right), \tag{8}$$

and from standard derivations for Gaussian we have that:

$$q(\mathbf{x}^{(\alpha_{t-1})}|\mathbf{x}^{(\alpha_t)}, \mathbf{x}^{(\alpha_0)}) = \mathcal{N}\left(\underbrace{\frac{\sqrt{\alpha_{t-1} - \alpha_t}}{1 - \alpha_t}\mathbf{x}^{(\alpha_0)} + \frac{\alpha_t(1 - \alpha_{t-1})}{\alpha_{t-1}(1 - \alpha_t)}\mathbf{x}^{(\alpha_t)}}_{\tilde{\mu}(\mathbf{x}^{(\alpha_t)}, \mathbf{x}^{(\alpha_0)}; \alpha_t, \alpha_{t-1})}, \frac{1 - \alpha_{t-1}}{1 - \alpha_t}\left(1 - \frac{\alpha_t}{\alpha_{t-1}}\right)\mathbf{I}\right). \tag{9}$$

As a variational approximation to the above, [42] considered a specific type of $p_\theta(\mathbf{x}^{(\alpha_{t-1})}|\mathbf{x}^{(\alpha_t)})$:

$$p_\theta(\mathbf{x}^{(\alpha_{t-1})}|\mathbf{x}^{(\alpha_t)}) = \mathcal{N}\left(\mu_\theta(\mathbf{x}^{(\alpha_t)}; \alpha_t, \alpha_{t-1}), (\sigma^{(\alpha_t)})^2\mathbf{I}\right), \tag{10}$$

where $\mu_\theta$ and $\sigma^{(\alpha_t)}$ are parameters, and we remove the superscript of $p_\theta$ to indicate that there are no additional discretization steps in between (the sampling process is explicitly defined). Then, we have the standard variational objective as follows:

$$L := \mathbb{E}_q\left[\log q(\mathbf{x}^{(\alpha_T)}|\mathbf{x}^{(\alpha_0)}) + \sum_{t=2}^T \log q(\mathbf{x}^{(\alpha_{t-1})}|\mathbf{x}^{(\alpha_t)}, \mathbf{x}^{(\alpha_0)}) - \sum_{t=1}^T \log p_\theta^{(\alpha_t, \alpha_{t-1})}(\mathbf{x}^{(\alpha_{t-1})}|\mathbf{x}^{(\alpha_t)})\right]$$

$$\equiv \mathbb{E}_q\left[\sum_{t=2}^T \underbrace{D_{\mathrm{KL}}(q(\mathbf{x}^{(\alpha_{t-1})}|\mathbf{x}^{(\alpha_t)}, \mathbf{x}^{(\alpha_0)}))\|p_\theta(\mathbf{x}^{(\alpha_{t-1})}|\mathbf{x}^{(\alpha_t)}))}_{L_{t-1}} - \log p_\theta(\mathbf{x}^{(\alpha_0)}|\mathbf{x}^{(\alpha_1)})\right],$$

where $\equiv$ denotes "equal up to a constant that does not depend on $\theta$" and each $L_{t-1}$ is a KL divergence between two Gaussian distributions. Let us assume that the standard deviation of $p_\theta(\mathbf{x}^{(\alpha_{t-1})}|\mathbf{x}^{(\alpha_t)})$ is equal to that of $q(\mathbf{x}^{(\alpha_{t-1})}|\mathbf{x}^{(\alpha_t)}, \mathbf{x}^{(\alpha_0)}))$, which we denote as $\sigma^{(\alpha_t)}$. And thus:

$$L_{t-1} = \mathbb{E}_q\left[\frac{1}{2(\sigma^{(\alpha_t)})^2}\|\mu_\theta(\mathbf{x}^{(\alpha_t)}; \alpha_t, \alpha_{t-1}) - \tilde{\mu}(\mathbf{x}^{(\alpha_t)}, \mathbf{x}^{(\alpha_0)}; \alpha_t, \alpha_{t-1})\|_2^2\right]. \tag{11}$$

With a particular reparametrization from $\mu_\theta$ to $\epsilon_\theta$ (which tries to model the noise vector at $\alpha_t$):

$$\mu_\theta(\mathbf{x}^{(\alpha_t)}; \alpha_t, \alpha_{t-1}) = \sqrt{\frac{\alpha_{t-1}}{\alpha_t}}\left(\mathbf{x}^{(\alpha_t)} - \frac{\sqrt{\alpha_{t-1} - \alpha_t}}{\sqrt{(1 - \alpha_t)\alpha_t}} \cdot \epsilon_\theta(\mathbf{x}^{(\alpha_t)}; \alpha_t)\right), \tag{12}$$

the objective function can be simplified to:

$$L_{t-1} = \mathbb{E}_{\mathbf{x}_0, \epsilon}\left[\frac{(\alpha_{t-1} - \alpha_t)}{2(\sigma^{(\alpha_t)})^2(1 - \alpha_t)\alpha_t}\|\epsilon - \epsilon_\theta(\mathbf{x}^{(\alpha_t)}; \alpha_t, \alpha_{t-1})\|_2^2\right] \tag{13}$$

where $\mathbf{x}^{(\alpha_t)} = \sqrt{\alpha_t}\mathbf{x}_0 + \sqrt{1 - \alpha_t}\epsilon$. Intuitively, this is a weighted sum of mean-square errors between the noise model $\epsilon_\theta$ and the actual noise $\epsilon$. Other weights can also be derived with different forward processes that are non-Markovian [84], and in practice, setting the weights to 1 is observed to achieve decent performance for image generation.

## A.2 DDIM sampling procedure

In this section, we discuss the detailed sampling procedure from $\mathbf{x}^{(0)} \sim \mathcal{N}(0, \boldsymbol{I})$ (which is the distribution with "all noise"[8]) to $\mathbf{x}^{(1)}$ (which is the model distribution with "no noise"). More specifically, we discuss a deterministic sampling procedure, which casts the generation procedure as an implicit model [84]. Compared to other procedures (such as the one in DDPM [42]), this has the advantage of better sample quality when few steps are allowed to produce each sample, as well as a near-invertible mapping between $\mathbf{x}^{(0)}$ and $\mathbf{x}^{(1)}$. We describe this procedure in Algorithm 2, where we can choose different series of $\alpha$ to control how many steps (and through which steps) we wish to draw a sample. The DDIM sampling procedure corresponds to a particular discretization to an ODE, we note that it is straightforward to also define the sampling procedure between any two $\alpha$ values. Similarly, given an observation $\mathbf{x}^{(1)}$ we can obtain the corresponding latent code $\mathbf{x}^{(0)}$ by sampling running Algorithm 2 with the sequence of $\alpha$ reversed.

---

**Algorithm 2** Sampling with the DDIM procedure

---

1: **Input**: non-increasing series $\{\alpha_t\}_{t=0}^T$ with $\alpha_T = 0$ and $\alpha_0 = 1$.
2: Sample $\mathbf{x}^{(1)} \sim \mathcal{N}(0, \boldsymbol{I})$.
3: **for** $k \leftarrow T$ to $1$ **do**
4:     Update $\mathbf{x}^{(\alpha_{t-1})}$ from $\mathbf{x}^{(\alpha_t)}$ such that

$$\sqrt{\frac{1}{\alpha_{t-1}}}\mathbf{x}^{(\alpha_{t-1})} = \sqrt{\frac{1}{\alpha_t}}\mathbf{x}^{(\alpha_t)} + \left( \sqrt{\frac{1 - \alpha_{t-1}}{\alpha_{t-1}}} - \sqrt{\frac{1 - \alpha_t}{\alpha_t}} \right) \cdot \epsilon_\theta(\mathbf{x}^{(\alpha_t)}; \alpha_t)$$

5: **end for**
6: **Output** $\mathbf{x}^{(0)}$.

---

## A.3 Contrastive representation learning

In contrastive representation learning, the goal is to distinguish a *positive* pair $(\mathbf{y}, \mathbf{w}) \sim p(\mathbf{y}, \mathbf{w})$ from $(m - 1)$ *negative* pairs $(\mathbf{y}, \overline{\mathbf{w}}) \sim p(\mathbf{y})p(\mathbf{w})$. In our context, the positive pairs are representations from the same image, and negative pairs are representations from different images; these images are pre-processed with strong data augmentations [14] to encourage rich representations. With two random, independent data augmentation procedures defined as $\mathrm{aug}_1$ and $\mathrm{aug}_2$, we define $p(\mathbf{y}, \mathbf{w})$ and $p(\mathbf{y})p(\mathbf{w})$ via the following sampling procedure:

$$(\mathbf{y}, \mathbf{w}) \sim p(\mathbf{y}, \mathbf{w}) : \mathbf{y} \sim q_\phi(\boldsymbol{z}^{(1)}|\mathrm{aug}_1(\mathbf{x})), \mathbf{w} \sim q_\phi(\boldsymbol{z}^{(1)}|\mathrm{aug}_2(\mathbf{x})), \mathbf{x} \sim p_{\mathrm{data}}(\mathbf{x}),$$

$$(\mathbf{y}, \mathbf{w}) \sim p(\mathbf{y})p(\mathbf{w}) : \mathbf{y} \sim q_\phi(\boldsymbol{z}^{(1)}|\mathrm{aug}_1(\mathbf{x}_1)), \mathbf{w} \sim q_\phi(\boldsymbol{z}^{(1)}|\mathrm{aug}_2(\mathbf{x}_2)), \mathbf{x}_1, \mathbf{x}_2 \sim p_{\mathrm{data}}(\mathbf{x}).$$

For a batch of $n$ positive pairs $\{(\mathbf{y}_i, \mathbf{w}_i)\}_{i=1}^n$, the contrastive predictive coding (CPC, [94]) objective is defined as:

$$L_{\mathrm{CPC}}(g; q_\phi) := \mathbb{E}\left[ \frac{1}{n} \sum_{i=1}^n \log \frac{m \cdot g(\mathbf{y}_i, \mathbf{w}_i)}{g(\mathbf{y}_i, \mathbf{w}_i) + \sum_{j=1}^{m-1} g(\mathbf{y}_i, \overline{\mathbf{w}_{i,j}})} \right] \tag{14}$$

for some positive critic function $g : \mathcal{Y} \times \mathcal{Z} \to \mathbb{R}_+$, where the expectation is taken over $n$ positive pairs $(\mathbf{y}_i, \mathbf{w}_i) \sim p(\mathbf{y}, \mathbf{w})$ and $n(m - 1)$ negative pairs $(\mathbf{y}_i, \overline{\mathbf{w}_{i,j}}) \sim p(\mathbf{y})p(\mathbf{w})$. Another interpretation to CPC is that it performs $m$-way classification where the ground truth label is assigned to the positive pair. The representation learner $q_\phi$ then aims to maximize the CPC objective, or to minimize the following objective:

$$-L_{\mathrm{C}}(q_\phi) := \min_g -L_{\mathrm{CPC}}(g; q_\phi), \tag{15}$$

Different specific implementations, such as MoCo [37, 16, 18] and SimCLR [14] can all be treated as specific implementations of this objective function. In this paper, we considered using MoCo-v2 [14] as our implementation for $L_{\mathrm{C}}$ objective; in principle, other implementations to CPC can also be integrated into D2C as well.

---

[8]Technically, the maximum noise level $\alpha_T$ should have $\alpha_T \to 0$ but not equal to 0, but we can approximate the distribution of $\mathbf{x}^{(\alpha_T)}$ with that of $\mathbf{x}^{(0)}$ arbitrarily well in practice.

## A.4 Training D2C

In Algorithm 3, we describe a high-level procedure that trains the D2C model; we note that this procedure does not have any adversarial components. On the high-level, this is the integration of three objectives: the reconstruction objective via the autoencoder, the diffusion objective over the latent space, and the contrastive objective over the latent space. In principle, the [reconstruction], [constrastive], and [diffusion] components can be optimized jointly or separately; we observe that normalizing the latent $\mathbf{z}^{(1)}$ with a global mean and standard deviation before applying the diffusion objective helps learning the diffusion model with a fixed $\alpha$ series.

---

**Algorithm 3** Training D2C

**Input**: Data distribution $p_{\text{data}}$.
**while** training **do**

**[Draw samples with data augmentation]**

Draw $m$ samples $\mathbf{x}_{0:m-1} \sim p_{\text{data}}(\mathbf{x})$.
Draw $(m+1)$ data augmentations $\text{aug}_0, \ldots \text{aug}_{m-1}$ and $\overline{\text{aug}}_0$.
**for** $i \leftarrow 0$ to $m-1$ **do**
 Draw $\mathbf{z}_i^{(1)} \sim q_\phi(\mathbf{z}^{(1)}|\text{aug}_i(\mathbf{x}))$.
**end for**
Draw $\overline{\mathbf{z}}_0^{(1)} \sim q_\phi(\mathbf{z}^{(1)}|\overline{\text{aug}}_0(\mathbf{x}))$.

**[Reconstruction]**

Reconstruct $\mathbf{x}_0 \sim p_\theta(\mathbf{x}|\mathbf{z}_0^{(1)})$
Minimize $L_{\text{recon}} = -\log p_\theta(\mathbf{x}|\mathbf{z}_0^{(1)})$ over $\theta$ and $\phi$ with gradient descent.

**[Contrastive]**

Define a classification task: assign label 1 to $(\mathbf{z}_0^{(1)}, \overline{\mathbf{z}}_0^{(1)})$ and label 0 to $(\mathbf{z}_0^{(1)}, \mathbf{z}_i^{(1)})$ for $i \neq 0$.
Define $L_{\text{CPC}}(g; q_\phi)$ as the loss to minimize for the above task, with $g$ as the classifier.
Define $\hat{g}$ as a minimizer to the classifier objective $L_{\text{CPC}}(g; q_\phi)$.
Minimize $L_{\text{CPC}}(\hat{g}; q_\phi)$ over $\phi$ with gradient descent.

**[Diffusion]**

Sample $\epsilon \sim \mathcal{N}(0, I)$, $t \sim \text{Uniform}(1, \ldots, T)$.
Define $\mathbf{z}_0^{(\alpha_t)} = \sqrt{\alpha_t}\mathbf{z}_0^{(0)} + \sqrt{1 - \alpha_t}\epsilon$.
Minimize $\|\epsilon - \epsilon_\theta(\mathbf{z}_0^{(\alpha_t)}; \alpha_t)\|_2^2$ over $\theta$ with gradient descent.

**end while**

---

## A.5 Few-shot conditional generation

In order to perform few-shot conditional generation, we need to implement line 4 in Algorithm 1, where an unnormalized (energy-based) model is defined over the representations. After we have defined the energy-based model, we implement a procedure to draw samples from this unnormalized model. We note that our approach (marked in teal boxes) is only one way of drawing valid samples, and not necessarily the optimal one. Furthermore, these implementations can also be done over the image space (which is the case for DDIM-I), which may costs more to compute than over the latent space since more layers are needed in a neural network to process it.

For generation from labels, we would define the energy-based model over latents as the product of two components: the first is the "prior" over $\mathbf{z}^{(1)}$ as defined by the diffusion model and the second is the "likliehood" of the label $\mathbf{c}$ being true given the latent variable $\mathbf{z}^{(1)}$. This places high energy values to the latent variables that are likely to occur under the diffusion prior (so generated images are likely to have high quality) as well as latent variables that have the label $\mathbf{c}$. To sample from this energy-based model, we perform a rejection sampling procedure, where we reject latent samples from the diffusion model that have low discrminator values. This procedure is describe in Algorithm 4.

---

**Algorithm 4** Generate from labels

**Input** model $r_\psi(\mathbf{c}|\mathbf{z}^{(1)})$, target label $\mathbf{c}$.

> **Define latent energy-based model**
>
> $E(\hat{\mathbf{z}}^{(1)}) = r_\psi(\mathbf{c}|\hat{\mathbf{z}}^{(1)}) \cdot p_\theta^{(1)}(\hat{\mathbf{z}}^{(1)})$

> **Sample from** $E(\hat{\mathbf{z}}^{(1)})$
>
> **while** True **do**
>     Sample $\hat{\mathbf{z}}^{(1)} \sim p_\theta^{(1)}(\hat{\mathbf{z}}^{(1)})$;
>     Sample $u \sim \text{Uniform}(0,1)$;
>     **If** $u < r_\psi(\mathbf{c}|\hat{\mathbf{z}}^{(1)})$ **then** break.
> **end while**

> **Output** $\hat{\mathbf{x}} \sim p_\theta(\mathbf{x}|\hat{\mathbf{z}}^{(1)})$.

---

For generation from manipulation constraints, we need to further define a prior that favors closeness to the given latent variable so that the manipulated generation is close to the given image except for the label $\mathbf{z}$. If the latent variable for the original image is $\mathbf{z}^{(1)} \sim q_\phi(\mathbf{z}^{(1)}|\mathbf{x})$, then we define the closeness via the L2 distance between the it and the manipulated latent. We obtain the energy-based model by multiplying this with the diffusion "prior" and the classifier "likelihood". Then, we approximately draw samples from this energy by taking a gradient step from the original latent value $\mathbf{z}^{(1)}$ and then regularizing it with the diffusion prior; this is described in Algorithm 5. A step size $\eta$, diffusion noise magnitude $\alpha$, and the diffusion steps from $\alpha$ to 1 are chosen as hyperparameters. We choose one $\eta$ for each attribute, $\alpha \approx 0.9$, and number of discretization steps to be 5[9]; we tried $\alpha \in [0.65, 0.9]$ and found that our results are not very sensitive to values within this range. We list the $\eta$ values for each attribute (details in Appendix C).

We note that a more principled approach is to take gradient with respect to the entire energy function (*e.g.*, for Langevin dynamics), where the gradient over the DDIM can be computed with instantaneous change-of-variables formula [13]; we observe that our current version is computationally efficient enough to perform well.

---

**Algorithm 5** Generate from manipulation constraints

**Input** model $r_\psi(\mathbf{c}|\mathbf{z}^{(1)})$, target label $\mathbf{c}$, original image $\mathbf{x}$.
Acquire latent $\mathbf{z}^{(1)} \sim q_\phi(\mathbf{z}^{(1)}|\mathbf{x})$;
Fit a model $r_\psi(\mathbf{c}|\mathbf{z}^{(1)})$ over $\{(\mathbf{z}_i^{(1)}, \mathbf{c}_i)\}_{i=1}^n$

> **Define latent energy-based model**
>
> $$E(\hat{\mathbf{z}}^{(1)}) = r_\psi(\mathbf{c}|\hat{\mathbf{z}}^{(1)}) \cdot p_\theta^{(1)}(\hat{\mathbf{z}}^{(1)}) \cdot \|\mathbf{z}^{(1)} - \hat{\mathbf{z}}^{(1)}\|_2^2$$

> **Sample from** $E(\hat{\mathbf{z}}^{(1)})$ **(approximate)**
>
> Choose hyperparameters $\eta > 0, \alpha \in (0,1)$.
> Take a gradient step $\bar{\mathbf{z}}^{(1)} \leftarrow \mathbf{z}^{(1)} + \eta \nabla_\mathbf{z} r_\psi(\mathbf{c}|\mathbf{z})|_{\mathbf{z}=\mathbf{z}^{(1)}}$.
> Add noise $\tilde{\mathbf{z}}^{(\alpha)} \leftarrow \sqrt{\alpha}\bar{\mathbf{z}}^{(1)} + \sqrt{1-\alpha}\epsilon$.
> Sample $\hat{\mathbf{z}}^{(1)} \sim p_\theta^{(\alpha,1)}(\mathbf{z}^{(1)}|\tilde{\mathbf{z}}^{(\alpha)})$ with DDIM, *i.e.*, use the diffusion prior to "denoise".

**Output** $\hat{\mathbf{x}} \sim p_\theta(\mathbf{x}|\hat{\mathbf{z}}^{(1)})$.

---

[9]The results are not particularly sensitive to how the discretization steps are chosen. For example, one can take $0.9 \rightarrow 0.92 \rightarrow 0.96 \rightarrow 0.98 \rightarrow 0.99 \rightarrow 1$.

# B Formal Statements and Proofs

## B.1 Relationship to maximum likelihood

**Theorem 1.** *(informal) For any valid $\{\alpha_i\}_{i=0}^T$, there exists some weights $\hat{w} : \{\alpha_i\}_{i=1}^T \to \mathbb{R}_+$ for the diffusion objective such that $-L_{\text{D2}}$ is a variational lower bound to the log-likelihood, i.e.,*

$$-L_{\text{D2}}(\theta, \phi; \hat{w}) \leq \mathbb{E}_{p_{\text{data}}}[\log p_\theta(\mathbf{x})], \tag{6}$$

*where $p_\theta(\mathbf{x}) := \mathbb{E}_{\mathbf{z}_0 \sim p^{(0)}(\mathbf{z}^{(0)})}[p_\theta(\mathbf{x}|\mathbf{z}^{(0)})]$ is the marginal probability of $\mathbf{x}$ under the D2C model.*

**Theorem 3.** *(formal) Suppose that $\mathbf{x} \in \mathbb{R}^d$. For any valid $\{\alpha_i\}_{i=0}^T$, let $\hat{w}$ satisfy:*

$$\forall t \in [2, \ldots, T], \quad \hat{w}(\alpha_t) = \frac{(1-\alpha_t)\alpha_{t-1}}{2(1-\alpha_{t-1})^2 \alpha_t} \tag{16}$$

$$\hat{w}(\alpha_1) = \frac{1-\alpha_1}{2(2\pi)^d \alpha_1} \tag{17}$$

*then:*

$$-L_{\text{D2}}(\theta, \phi; \hat{w}) + H(q_\phi(\mathbf{z}^{(1)}|\mathbf{x})) \leq \mathbb{E}_{p_{\text{data}}}[\log p_\theta(\mathbf{x})] \tag{18}$$

*where $p_\theta(\mathbf{x}) := \mathbb{E}_{\mathbf{x}_0 \sim p^{(0)}(\mathbf{z}^{(0)})}[p_\theta(\mathbf{x}|\mathbf{z}^{(0)})]$ is the marginal probability of $\mathbf{x}$ under the D2C model.*

*Proof.* First, we have that:

$$\mathbb{E}_{p_{\text{data}}(\mathbf{x})}[\log p_\theta(\mathbf{x})] = \mathbb{E}_{p_{\text{data}}(\mathbf{x})}\left[\log \sum_{\mathbf{z}^{(1)}} p_\theta(\mathbf{x}|\mathbf{z}^{(1)}) p_\theta(\mathbf{z}^{(1)})\right] \tag{19}$$

$$\geq \mathbb{E}_{p_{\text{data}}(\mathbf{x}), q_\phi(\mathbf{z}^{(1)})}[\log p_\theta(\mathbf{x}|\mathbf{z}^{(1)}) + \log p_\theta(\mathbf{z}^{(1)}) - \log q_\phi(\mathbf{z}^{(1)}|\mathbf{x})] \tag{20}$$

$$= \mathbb{E}_{p_{\text{data}}(\mathbf{x}), q_\phi(\mathbf{z}^{(1)}|\mathbf{x})}[\log p_\theta(\mathbf{x}|\mathbf{z}^{(1)}) - D_{\text{KL}}(q_\phi(\mathbf{z}^{(1)}|\mathbf{x}) \| p_\theta(\mathbf{z}^{(1)}))]. \tag{21}$$

where we use Jensen's inequality here. Compared with the objective for D2:

$$-L_{\text{D2}}(\theta, \phi; w) := \mathbb{E}_{\mathbf{x} \sim p_{\text{data}}, \mathbf{z}^{(1)} \sim q_\phi(\mathbf{z}^{(1)}|\mathbf{x})}[\log p(\mathbf{x}|\mathbf{z}^{(1)}) - \ell_{\text{diff}}(\mathbf{z}^{(1)}; w, \theta)], \tag{22}$$

and it is clear the proof is complete if we show that:

$$H(q_\phi(\mathbf{z}^{(1)}|\mathbf{x})) - \mathbb{E}_{\mathbf{z}^{(1)} \sim q_\phi(\mathbf{z}^{(1)}|\mathbf{x})}[\ell_{\text{diff}}(\mathbf{z}^{(1)}; \hat{w}, \theta)] \tag{23}$$

$$\leq -D_{\text{KL}}(q_\phi(\mathbf{z}^{(1)}|\mathbf{x}) \| p_\theta(\mathbf{z}^{(1)})) \tag{24}$$

$$= H(q_\phi(\mathbf{z}^{(1)}|\mathbf{x})) + \mathbb{E}_{\mathbf{z}^{(1)} \sim q_\phi(\mathbf{z}^{(1)}|\mathbf{x})}[\log p_\theta(\mathbf{z}^{(1)})] \tag{25}$$

or equivalently:

$$\mathbb{E}_{\mathbf{z}^{(1)} \sim q_\phi(\mathbf{z}^{(1)}|\mathbf{x})}[\ell_{\text{diff}}(\mathbf{z}^{(1)}; \hat{w}, \theta)] \leq \mathbb{E}_{\mathbf{z}^{(1)} \sim q_\phi(\mathbf{z}^{(1)}|\mathbf{x})}[\log p_\theta(\mathbf{z}^{(1)})] \tag{26}$$

Let us apply variational inference with an inference model $q(\mathbf{z}^{(\alpha_{1:T})}|\mathbf{z}^{(1)})$ where $\alpha_0 = 1$ and $\alpha_T = 0$:

$$\mathbb{E}_{\mathbf{z}^{(1)} \sim q_\phi(\mathbf{z}^{(1)}|\mathbf{x})}[\log p_\theta(\mathbf{z}^{(1)})] = \mathbb{E}_{\mathbf{z}^{(1)} \sim q_\phi(\mathbf{z}^{(1)}|\mathbf{x})}[\log \sum_{\mathbf{z}} \left(p_\theta(\mathbf{z}^{(\alpha_T)}) \prod_{t=1}^T p_\theta(\mathbf{z}^{(\alpha_{t-1})}|\mathbf{z}^{(\alpha_t)})\right)]$$

$$\geq \mathbb{E}_{\mathbf{z}^{(\alpha_{0:T})}}[\log p_\theta(\mathbf{z}^{(\alpha_T)}) + \sum_{t=1}^T \log p_\theta(\mathbf{z}^{(\alpha_{t-1})}|\mathbf{z}^{(\alpha_t)}) - \log q(\mathbf{z}^{(\alpha_{1:T})}|\mathbf{z}^{(\alpha_0)})] \tag{27}$$

$$\geq \mathbb{E}_{\mathbf{z}^{(\alpha_{0:T})}}\Big[\log p_\theta(\mathbf{z}^{(\alpha_T)}) - \log q(\mathbf{z}^{(\alpha_T)}|\mathbf{z}^{(\alpha_0)}) \tag{28}$$

$$- \sum_{t=2}^T \underbrace{D_{\text{KL}}(q(\mathbf{z}^{(\alpha_{t-1})}|\mathbf{z}^{(\alpha_t)}, \mathbf{z}^{(\alpha_0)}) \| p_\theta(\mathbf{z}^{(\alpha_{t-1})}|\mathbf{z}^{(\alpha_t)}))}_{L_{t-1}} + \log p_\theta(\mathbf{z}^{(\alpha_0)}|\mathbf{z}^{(\alpha_1)})\Big]$$

where we remove the superscript of $p_\theta$ to indicate that there are no intermediate discretization steps between $\alpha_{t-1}$ and $\alpha_t$. Now, for $t \geq 2$, let us consider $p_\theta$ and $q_\phi$ with the form in Equations 9

and 10 respectively, which are both Gaussian distributions (restrictions to $p_\theta$ will still give lower bounds). Then we can model the standard deviation of $p_\theta(\mathbf{x}^{(\alpha_{t-1})}|\mathbf{x}^{(\alpha_t)})$ to be equal to that of $q(\mathbf{x}^{(\alpha_{t-1})}|\mathbf{x}^{(\alpha_t)}, \mathbf{x}^{(\alpha_0)}))$. Under this formulation, the KL divergence for $L_{t-1}$ is just one between two Gaussians with the same standard deviations and is a weighted Euclidean distance between the means. Using the derivation from Equation (11) to Equation (13), we have that:

$$L_{t-1} = \mathbb{E}_{\mathbf{z}_0, \epsilon} \left[ \frac{(1-\alpha_t)\alpha_{t-1}}{2(1-\alpha_{t-1})^2 \alpha_t} \|\epsilon - \epsilon_\theta(\mathbf{z}^{(\alpha_t)}; \alpha_t, \alpha_{t-1})\|_2^2 \right] \tag{29}$$

which gives us the weights for $\hat{w}$ for $\alpha_{2:T}$. For $p_\theta(\mathbf{z}^{(\alpha_0)}|\mathbf{z}^{(\alpha_1)})$ let us model it to be a Gaussian with mean

$$\mu_\theta(\mathbf{z}^{(\alpha_1)}; \alpha_1, \alpha_0) = \frac{\mathbf{z}^{(\alpha_1)} - \sqrt{1-\alpha_t}\epsilon_\theta(\mathbf{z}^{(\alpha_1)}; \alpha_1, \alpha_0)}{\sqrt{\alpha_1}}$$

and standard deviation $1/\sqrt{2\pi}$ (chosen such that normalization constant is 1). Thus, with

$$\mathbf{z}^{(0)} = \frac{\mathbf{z}^{(\alpha_1)} - \sqrt{1-\alpha_t}\epsilon}{\sqrt{\alpha_1}}$$

we have that:

$$\log p_\theta(\mathbf{z}^{(\alpha_0)}|\mathbf{z}^{(\alpha_1)}) = \frac{1-\alpha_1}{2(2\pi)^d \alpha_1} \|\epsilon - \epsilon_\theta(\mathbf{z}^{(\alpha_1)}; \alpha_1, \alpha_0)\|_2^2 \tag{30}$$

which gives us the weight of $\hat{w}$ for $\alpha_1$. Furthermore:

$$\mathbb{E}_{\mathbf{z}^{(\alpha_{0:T})}}[\log p_\theta(\mathbf{z}^{(\alpha_T)}) - q(\mathbf{z}^{(\alpha_T)}|\mathbf{z}^{(\alpha_0)})] = 0 \tag{31}$$

because $\mathbf{z}^{(\alpha_T)} \sim \mathcal{N}(0, \boldsymbol{I})$ for both $p_\theta$ and $q$. Therefore, we have that:

$$\mathbb{E}_{\mathbf{z}^{(1)} \sim q_\phi(\mathbf{z}^{(1)}|\mathbf{x})}[\ell_{\text{diff}}(\mathbf{z}^{(1)}; \hat{w}, \theta)] \leq \mathbb{E}_{\mathbf{z}^{(1)} \sim q_\phi(\mathbf{z}^{(1)}|\mathbf{x})}[\log p_\theta(\mathbf{z}^{(1)})] \tag{32}$$

which completes the proof. $\square$

## B.2 D2 models address latent posterior mismatch in VAEs

**Theorem 2.** *(informal) Let $p_\theta(\mathbf{z}) = \mathcal{N}(0, 1)$. For any $\epsilon > 0$, there exists a $q_\phi(\mathbf{z})$ with an $(\epsilon, 0.49)$-prior hole, such that $D_{\text{KL}}(q_\phi\|p_\theta) \leq \log 2^{10}$ and $W_2(q_\phi, p_\theta) < \gamma$ for any $\gamma > 0$, where $W_2$ is the 2-Wasserstein distance.*

**Theorem 4.** *(formal) Let $p_\theta(\mathbf{z}) = \mathcal{N}(0, \boldsymbol{I})$ where $\mathbf{z} \in \mathbb{R}^d$. For any $\epsilon > 0, \delta < 0.5$, there exists a distribution $q_\phi(\mathbf{z})$ with an $(\epsilon, \delta)$-prior hole, such that $D_{\text{KL}}(q_\phi\|p_\theta) \leq \log 2$ and $W_2(q_\phi, p_\theta) < \gamma$ for any $\gamma > 0$, where $W_2$ is the 2-Wasserstein distance.*

*Proof.* Let us define a function $f : \mathbb{R}_{\geq 0} \to [0, 1]$ such that for any Euclidean ball $B(0, R)$ centered at 0 with radius $R$:

$$f(R) := \int_{B(0,R)} p_\theta(\mathbf{z}) \mathrm{d}\mathbf{z}, \tag{33}$$

*i.e.*, $f(R)$ measures the probability mass of the Gaussian distribution $p_\theta(\mathbf{z})$ within $B(0, R)$. As $\mathrm{d}f/\mathrm{d}R > 0$ for $R > 0$, $f$ is invertible.

Now we shall construct $q_\phi(\mathbf{z})$. First, let $q_\phi(\mathbf{z}) = p_\theta(\mathbf{z})$ whenever $\|\mathbf{z}\|_2 \geq f^{-1}(2\delta)$; then for any $n$, we can find a sequence $\{r_0, r_1, \ldots, r_{2n}\}$ such that:

$$r_0 = 0, \quad r_{2n} = f^{-1}(2\delta), \quad f(r_i) - f(r_{i-1}) = f^{-1}(2\delta)\delta/n \text{ for all } k \in \{1, \ldots, 2n\}, \tag{34}$$

Intuitively, we find $2n$ circles with radii $\{r_0, \ldots, r_{2n}\}$ whose masses measured by $p_\theta(\mathbf{z})$ is an arithmetic progression $\{0, \delta/2n, \ldots, 2\delta\}$. We then define $q_\phi(\mathbf{z})$ for $\|\mathbf{z}\| < f^{-1}(2\delta)$ as follows:

$$q_\phi(\mathbf{z}) = \begin{cases} 2 \cdot p_\theta(\mathbf{z}) & \text{if } \|\mathbf{z}\| \in \bigcup_{k=0}^{n-1}[r_{2k}, r_{2k+1}) \\ 0 & \text{otherwise} \end{cases} \tag{35}$$

---

[10]This is reasonably low for realistic VAE models (NVAE [91] reports a KL divergence of around 2810 nats).

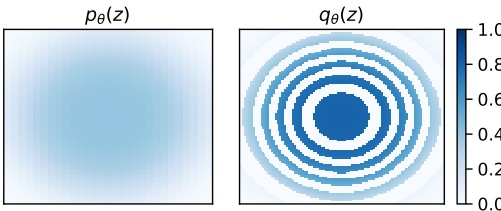

Figure 7: Illustration of the construction in 2d. When we use more rings, the prior hole and upper bound of KL divergence are constant but the upper bound of Wasserstein distance decreases.

Intuitively, $q_\phi$ is defined by moving all the mass from ring $(2k+1)$ to ring $2k$. Note that this $q_\phi(\mathbf{z})$ is a valid probability distribution because:

$$\int_{\mathbb{R}^d} q_\phi(\mathbf{z})\mathrm{d}\mathbf{z} = \int_{B(0,f^{-1}(2\delta))} q_\phi(\mathbf{z})\mathrm{d}\mathbf{z} + \int_{B^c(0,f^{-1}(2\delta))} q_\phi(\mathbf{z})\mathrm{d}\mathbf{z} \tag{36}$$

$$= 2\int_{B(0,f^{-1}(2\delta))} p_\theta(\mathbf{z})\mathbb{I}\left(\|\mathbf{z}\| \in \bigcup_{i=0}^{n-1}[r_{2k}, r_{2k+1})\right)\mathrm{d}\mathbf{z} + \int_{B^c(0,f^{-1}(2\delta))} p_\theta(\mathbf{z})\mathrm{d}\mathbf{z} \tag{37}$$

$$= \int_{B(0,f^{-1}(2\delta))} p_\theta(\mathbf{z})\mathrm{d}\mathbf{z} + \int_{B^c(0,f^{-1}(2\delta))} p_\theta(\mathbf{z})\mathrm{d}\mathbf{z} = 1 \tag{38}$$

Next, we validate that $q_\phi$ satisfies our constraints in the statement.

**Prior hole**  Apparently, if we choose $\mathcal{S} = \bigcup_{k=0}^{n-1}[r_{2k+1}, r_{2k+2})$, then $\int_\mathcal{S} p_\theta(\mathbf{z})\mathrm{d}\mathbf{z} = \delta$ and $\int_\mathcal{S} q_\phi(\mathbf{z})\mathrm{d}\mathbf{z} = 0$; so $\mathcal{S}$ instantiates a $(\epsilon, \delta)$-prior hole.

**KL divergence**  We note that $q_\phi(\mathbf{z}) \le 2p_\theta(\mathbf{z})$ is true for all $\mathbf{z}$, so
$$D_{\mathrm{KL}}(q_\phi(\mathbf{z})\|p_\theta(\mathbf{z})) = \mathbb{E}_{\mathbf{z}\sim q_\phi(\mathbf{z})}[\log q_\phi(\mathbf{z}) - \log p_\theta(\mathbf{z})] \le \log 2.$$

**2 Wasserstein Distance**  We use the Monge formulation:

$$W_2(q_\phi(\mathbf{z}), 2p_\theta(\mathbf{z})) = \min_{T:q_\phi=T_\sharp p_\theta} \int_{R^d} \|\mathbf{z} - T(\mathbf{z})\|_2^2 p_\theta(\mathbf{z})\mathrm{d}\mathbf{z} \tag{39}$$

where $T$ is any transport map from $p_\theta$ to $q_\phi$. Consider the transport map $\hat{T}$ such that:

$$\hat{T}(\mathbf{z}) = \begin{cases} \mathbf{z} & \text{if} \quad q_\phi(\mathbf{z}) \ge 0 \\ \mathbf{z} \cdot f^{-1}(f(\|\mathbf{z}\|) - f(r_{2k+1}) + k\delta/n) & \text{otherwise, for } k \text{ such that } \|\mathbf{z}\|_2 \le [r_{2k+1}, r_{2k+2}) \end{cases} \tag{40}$$

which moves the mass in $[r_{2k+1}, r_{2k+2})$ to $[r_{2k}, r_{2k+1})$. From this definition, we have that $\|\hat{T}(\mathbf{z}) - \mathbf{z}\|_2 \le \max_{k\in\{0,\dots,n-1\}}(r_{2k+2} - r_{2k})$. Moreover, since by definition,

$$2\delta/n = \int_{B(0,r_{2k+2})} p_\theta(\mathbf{z})\mathrm{d}\mathbf{z} - \int_{B(0,r_{2k})} p_\theta(\mathbf{z})\mathrm{d}\mathbf{z} \tag{41}$$

$$> \pi(r_{2k+2}^2 - r_{2k}^2)\min_{\mathbf{z}:\|\mathbf{z}\|\in[r_{2k},r_{2k+2})} p_\theta(\mathbf{z}) \tag{42}$$

$$> \pi(r_{2k+2} - r_{2k})^2\min_{\mathbf{z}:\|\mathbf{z}\|\in[r_{2k},r_{2k+2})} p_\theta(\mathbf{z}) \tag{43}$$

We have that

$$W_2(q_\phi(\mathbf{z}), 2p_\theta(\mathbf{z})) \le \max_{k\in\{0,\dots,n-1\}}(r_{2k+2} - r_{2k})^2 < \frac{2\delta}{\pi n \min_{\mathbf{z}:\|\mathbf{z}\|_2 \le r_{2n}} p_\theta(\mathbf{z})} \tag{44}$$

$$< \frac{2\delta}{\pi n \min_{\mathbf{z}:\|\mathbf{z}\|_2 \le r_{2n}} p_\theta(f^{-1}(2\delta)\mathbf{n})} \tag{45}$$

for any vector $\mathbf{n}$ with norm 1. Note that the above inequality is inversely proportional to $n$, which can be any integer. Therefore, for a fixed $\delta$, $W_2(q_\phi(\mathbf{z}), 2p_\theta(\mathbf{z})) = O(1/n)$; so for any $\gamma$, there exists $n$ such that $W_2(q_\phi(\mathbf{z}), 2p_\theta(\mathbf{z})) < \gamma$, completing the proof. $\qquad\square$

Table 5: Hyperparameters across different datasets

| Hyperparameter | CIFAR-10 32x32 | CIFAR-100 32x32 | CelebA-64 64x64 | fMoW 64x64 | CelebA-HQ-256 256x256 | FFHQ-256 256x256 |
|---|---|---|---|---|---|---|
| # of epochs | 1000 | 1000 | 300 | 300 | 200 | 100 |
| batch size per GPU | 32 | 32 | 16 | 16 | 3 | 3 |
| # initial channels in enc, | 128 | 128 | 64 | 64 | 24 | 24 |
| spatial dims of z | 16*16 | 16*16 | 32*32 | 32*32 | 64*64 | 64*64 |
| # channel in z | 8 | 8 | 5 | 5 | 8 | 8 |
| MoCo-v2 queue size | 65536 | 65536 | 65536 | 65536 | 15000 | 15000 |
| Diffusion feature map res. | 16,8,4,2 | 16,8,4,2 | 32,16,8,4,1 | 32,16,8,4,1 | 64,32,16,8,2 | 64,32,16,8,2 |
| $\lambda^{-1}$ | 17500 | 17500 | 17500 | 17500 | 17500 | 17500 |
| learning rate | 0.001 | 0.001 | 0.001 | 0.001 | 0.001 | 0.001 |
| Optimizer | AdamW | AdamW | AdamW | AdamW | AdamW | AdamW |
| # GPUs | 8 | 8 | 4 | 4 | 8 | 8 |
| GPU Type | 16 GB V100 | 16 GB V100 | 12 GB Titan X | 12 GB Titan X | 16 GB V100 | 16 GB V100 |
| Total training time (h) | 24 | 24 | 120 | 120 | 96 | 96 |

**Note on DDIM prior preventing the prior hole**    For a noise level $\alpha$, we have that:

$$q^{(\alpha)}(\mathbf{z}^{(\alpha)}) = \mathbb{E}_{\mathbf{z}^{(1)} \sim q^{(1)}(\mathbf{z}^{(1)})}[\mathcal{N}(\sqrt{\alpha}\mathbf{z}^{(1)}, (1-\alpha)\boldsymbol{I})] \tag{46}$$

as $\alpha \to 0$, $D_{\mathrm{KL}}(q^{(\alpha)}(\mathbf{z}^{(\alpha)})\|\mathcal{N}(0, \boldsymbol{I})) \to 0$. From Pinsker's inequality and the definition of $(\epsilon, \delta)$-prior hole:

$$\delta - \epsilon \le D_{\mathrm{TV}}(q^{(\alpha)}(\mathbf{z}^{(\alpha)}), \mathcal{N}(0, \boldsymbol{I}))) \le \sqrt{\frac{1}{2} D_{\mathrm{KL}}(q^{(\alpha)}(\mathbf{z}^{(\alpha)})\|\mathcal{N}(0, \boldsymbol{I}))}, \tag{47}$$

we should not expect to see any $(\epsilon, \delta)$-prior hole where the difference between $\delta$ and $\epsilon$ is large.

# C    Experimental details

## C.1    Architecture details and hyperparameters used for training

We modify the NVAE [91] architecture by removing the "Combiner Cells" in both encoder and decoder. For the diffusion model, we use the same architecture with different number of channel multiplications, as used in [42, 84]. For Contrastive learning, we use the MoCo-v2 [16] algorithm with augmentations such as *RandomResizedCrop, ColorJitter, RandomGrayscale, RandomHorizontalFlip*.

Additional details about the hyperparameters used are provided in Table 5.

## C.2    Additional details for conditional generation

For $r_\psi(\mathbf{c}|\mathbf{z}^{(1)})$ we consider training a linear model over the latent space, which has the advantage of being computationally efficient. For conditional generation on labels, we reject samples if their classifier return are lower than a certain threshold (we used $0.5$ for all our experiments). For conditional image manipulation, we consider the same step size $\eta$ for each attribute: $\eta = 10$ for *red lipstick* and $\eta = 15$ for *blond*. We note that these values are not necessarily the optimal ones, as the intensity of the change can grow with a choice of larger $\eta$ values.

## C.3    Amazon Mechanical Turk procedure

The mechanical turk evaluation is done for different attributes to find out how evaluators evaluate the different approaches. The evaluators are asked to compare a pair of images, and find the best image, which retains the identity as well as contains the desired attribute. Figure 8 a) shows the instructions that was given to the evaluators before starting the test and Figure 8 b) contains the UI shown to the evaluators when doing comparison. Each evaluation task contains 10 pairwise comparisons, and we perform 15 such evaluation tasks for each attribute. The reward per task is kept as 0.25$. Since each task takes around 2.5 mins, so the hourly wage comes to be 6$ per hour.

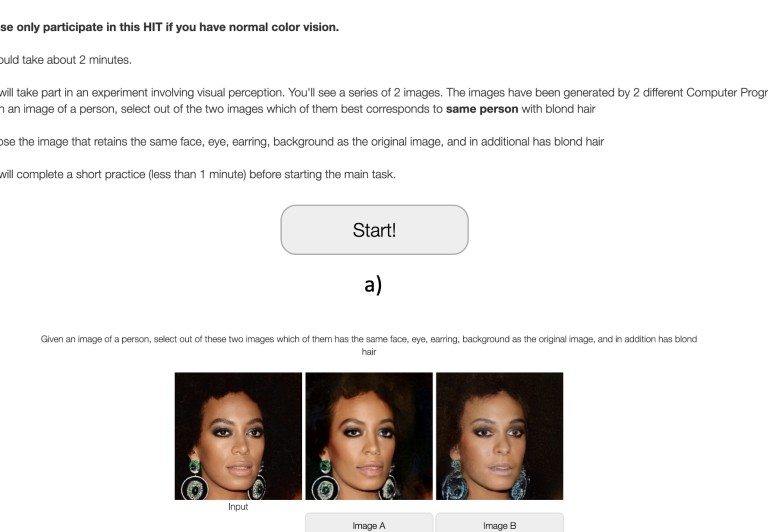

**About this HIT:**

- **Please only participate in this HIT if you have normal color vision.**
- It should take about 2 minutes.
- You will take part in an experiment involving visual perception. You'll see a series of 2 images. The images have been generated by 2 different Computer Programs. Given an image of a person, select out of the two images which of them best corresponds to **same person** with blond hair
- Choose the image that retains the same face, eye, earring, background as the original image, and in additional has blond hair
- You will complete a short practice (less than 1 minute) before starting the main task.

Start!

a)

Given an image of a person, select out of these two images which of them has the same face, eye, earring, background as the original image, and in addition has blond hair

Input | Image A | Image B

b)

Figure 8: a) Instructions shown to human evaluators for Amazon Mechanical Turk for blond hair before starting the evaluation and b) UI shown to the evaluators when doing comparison.

# D  Additional Results

**Sample quality versus speed**    It is known that the in diffusion models, one could achieve higher sample qualities by employing additional intermediate steps. In Table 6, we compare generation performance with DDPM and DDIM when we take 10, 50, and 100 steps to produce a sample.

Table 6: Sample quality as a function of diffusion steps.

| | CIFAR-10 | | | CIFAR-100 | | | CelebA-64 | | |
|---|---|---|---|---|---|---|---|---|---|
| Steps | 10 | 50 | 100 | 10 | 50 | 100 | 10 | 50 | 100 |
| DDPM [42] | 41.07 | 8.01 | 5.78 | 50.27 | 21.37 | 16.72 | 33.12 | 18.48 | 13.93 |
| DDIM [84] | **13.36** | **4.67** | **4.16** | **23.34** | **11.69** | **10.16** | 17.33 | 9.17 | 6.53 |
| D2 (Ours) | 22.3 | 15.8 | 15.1 | 28.35 | 19.81 | 19.85 | - | - | - |
| D2C (Ours) | 17.71 | 10.11 | 10.15 | 23.16 | 14.62 | 14.46 | **17.32** | **6.8** | **5.7** |

**CIFAR-10 image generation**    We list results for unconditional CIFAR-10 image generation for various types of generative models in Table 7. While our results are slightly worse than state-of-the-art diffusion models, we note that our D2C models are trained with relatively fewer resources that some of the baselines; for example, our D2C models is trained on 8 GPUs for 24 hours, whereas NVAE is trained on 8 GPUs for 100 hours and DDPM is trained on v3-8 TPUs for 24 hours. We also note that these comparisons are not necessarily fair in terms of the architecture and compute used to produce the samples.

Table 7: CIFAR-10 image generation results.

| Method | FID |
|---|---|
| NVAE [91] | 51.71 |
| NCP-VAE [3] | 24.08 |
| EBM [31] | 40.58 |
| StyleGAN2 [50] | 3.26 |
| DDPM [42] | 3.17 |
| DDIM [84] | 4.04 |
| D2C | 10.15 |

**Additional image generation results**    We list additional image generation results in Figure 9 (unconditional), Figures 10, 11, 12, and 13 (conditional on manipulation constraints), and Figures 14, 15, 16, and 17 (conditional on labels).

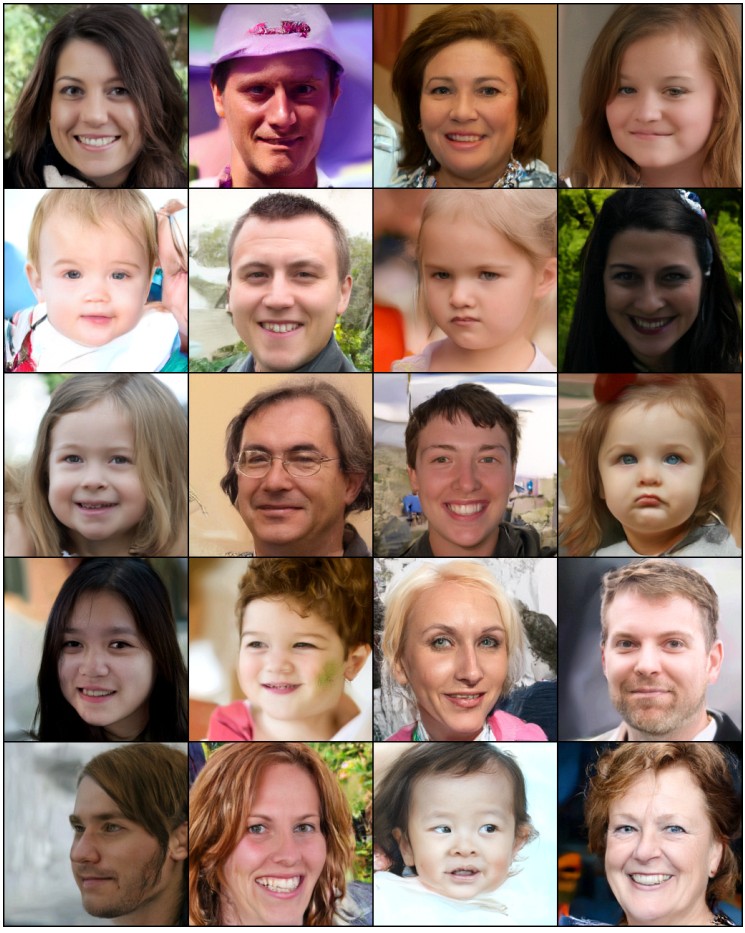

Figure 9: Additional image samples for the FFHQ-256 dataset.

**Reconstructing contrastive features**   Given the success of contrastive features in self-supervised learning, it might be possible that these representations alone can be good enough for reconstructing images. To see if this is true, we used a pre-trained MoCo-v2 model and trained a NVAE decoder to reconstruct the image. The reconstruction MSE per image was 58.20, significantly worse than NVAE (0.25) and D2C (0.76). Thus, the representations from the MoCo-v2 model are not necessarily well suited for generative modeling.

## E   Broader Impact

Recent approaches have trained large vision and language models for conditional generation [74]. However, training such models (*e.g.*, text to image generation) would require vast amounts of resources including data, compute and energy. Our work investigate ideas towards reducing the need to provide paired data (*e.g.*, image-text pairs) and instead focus on using unsupervised data.

Since our generative model tries to faithfully reconstruct training images, there is a potential danger that the model will inherit or exacerbate the bias within the data collection process [83]. Our method also has the risk of being used in unwanted scenarios such as deep fake. Nevertheless, if we are able to monitor and control how the latent variables are used in the downstream task (which may be easier than directly over images, as the latent variables themselves have rich structure), we can better defend against unwanted use of our models by rejecting problematic latent variables before decoding.

| Original | D2C | StyleGAN2 | NVAE | DDIM |
|----------|-----|-----------|------|------|

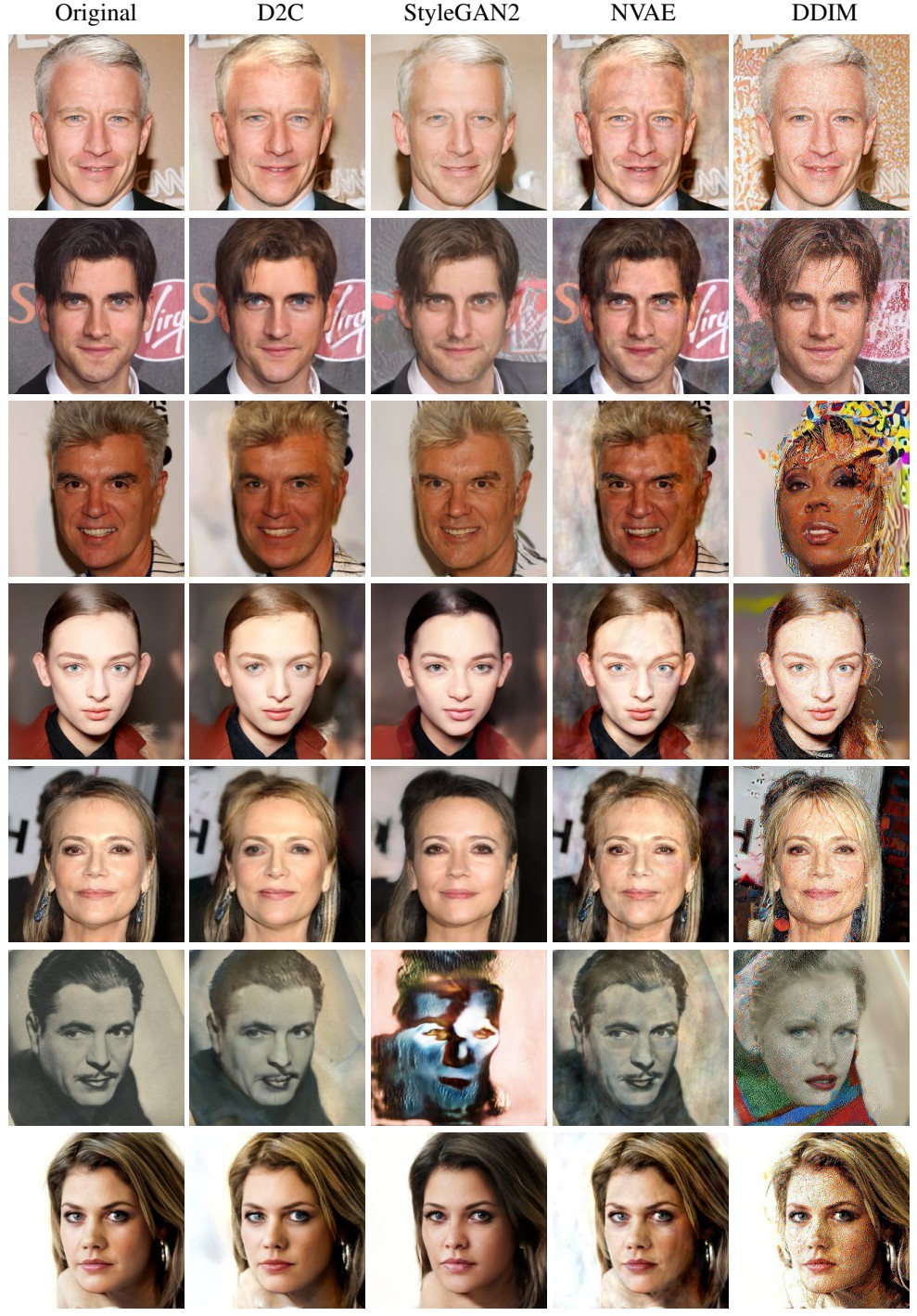

Figure 10: Image manipulation results for *blond hair*. More results can be found in `https://d2c-model.github.io/blond_png.html`.

Original D2C StyleGAN2 NVAE DDIM

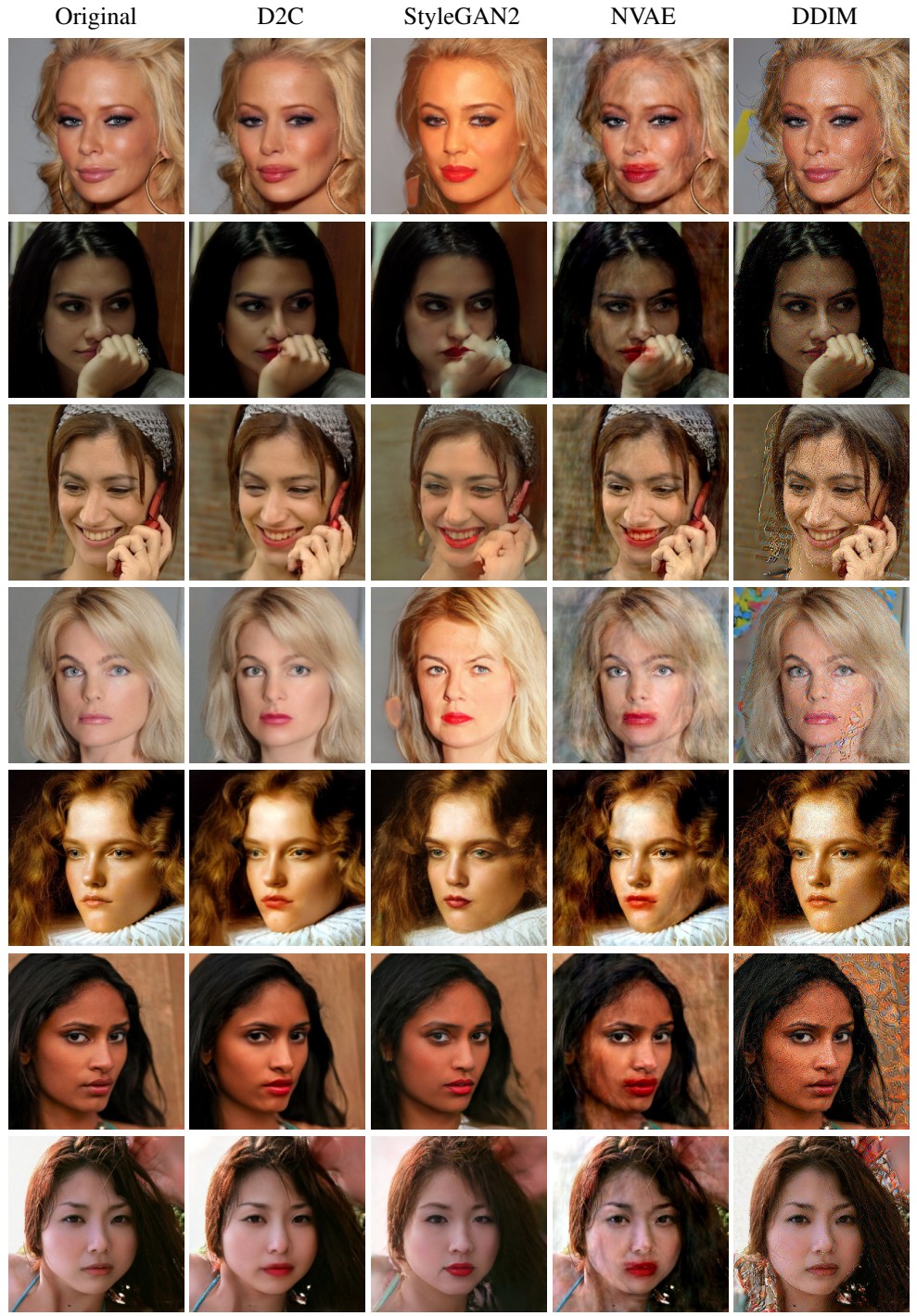

Figure 11: Image manipulation results for *red lipstick*. More results can be found in `https://d2c-model.github.io/red_lipstick_png.html`.

Original       D2C       StyleGAN2       NVAE       DDIM

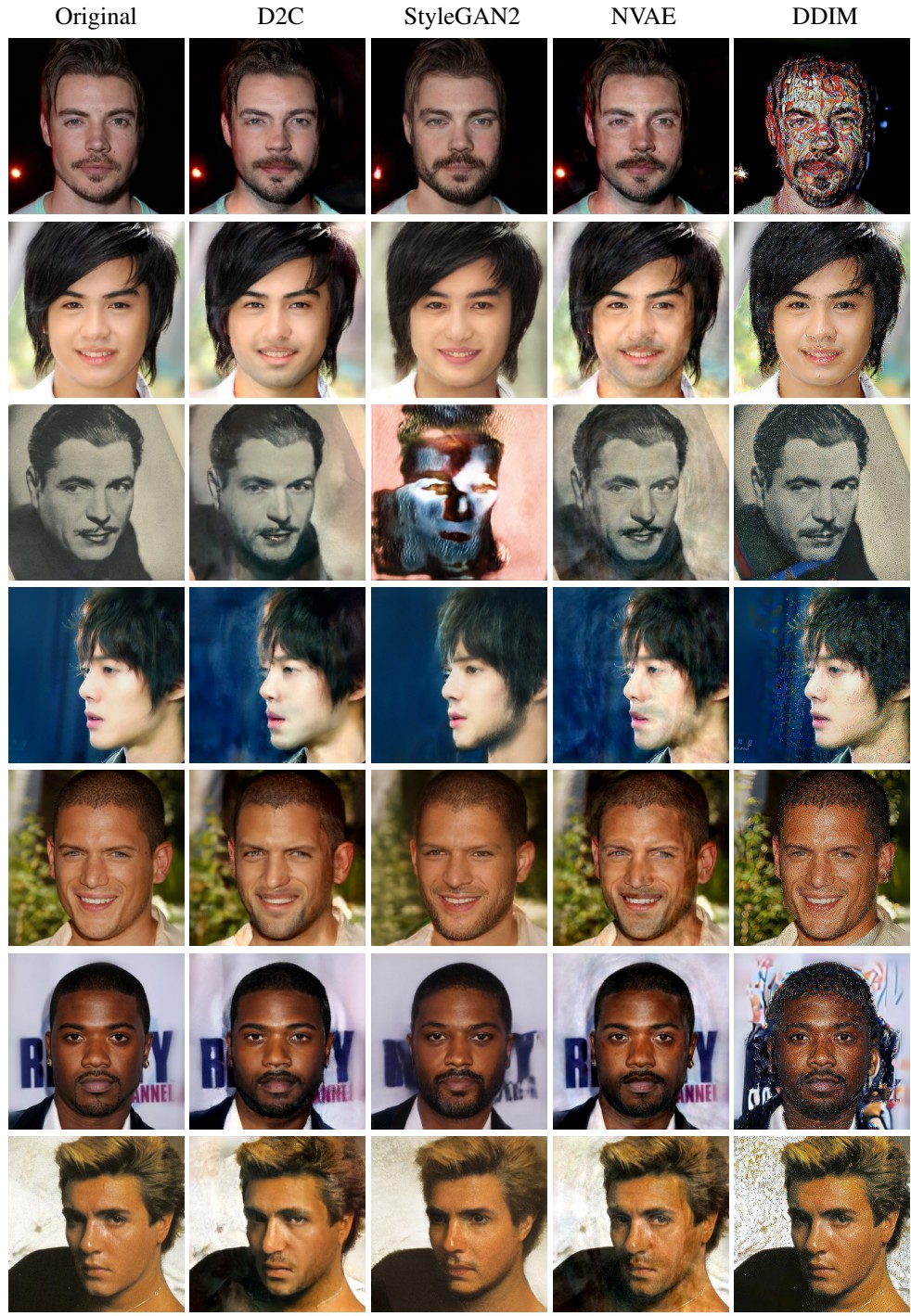

Figure 12: Image manipulation results for *beard*. More results can be found in https://d2c-model.github.io/beard_png.html.

Original  D2C  StyleGAN2  NVAE  DDIM

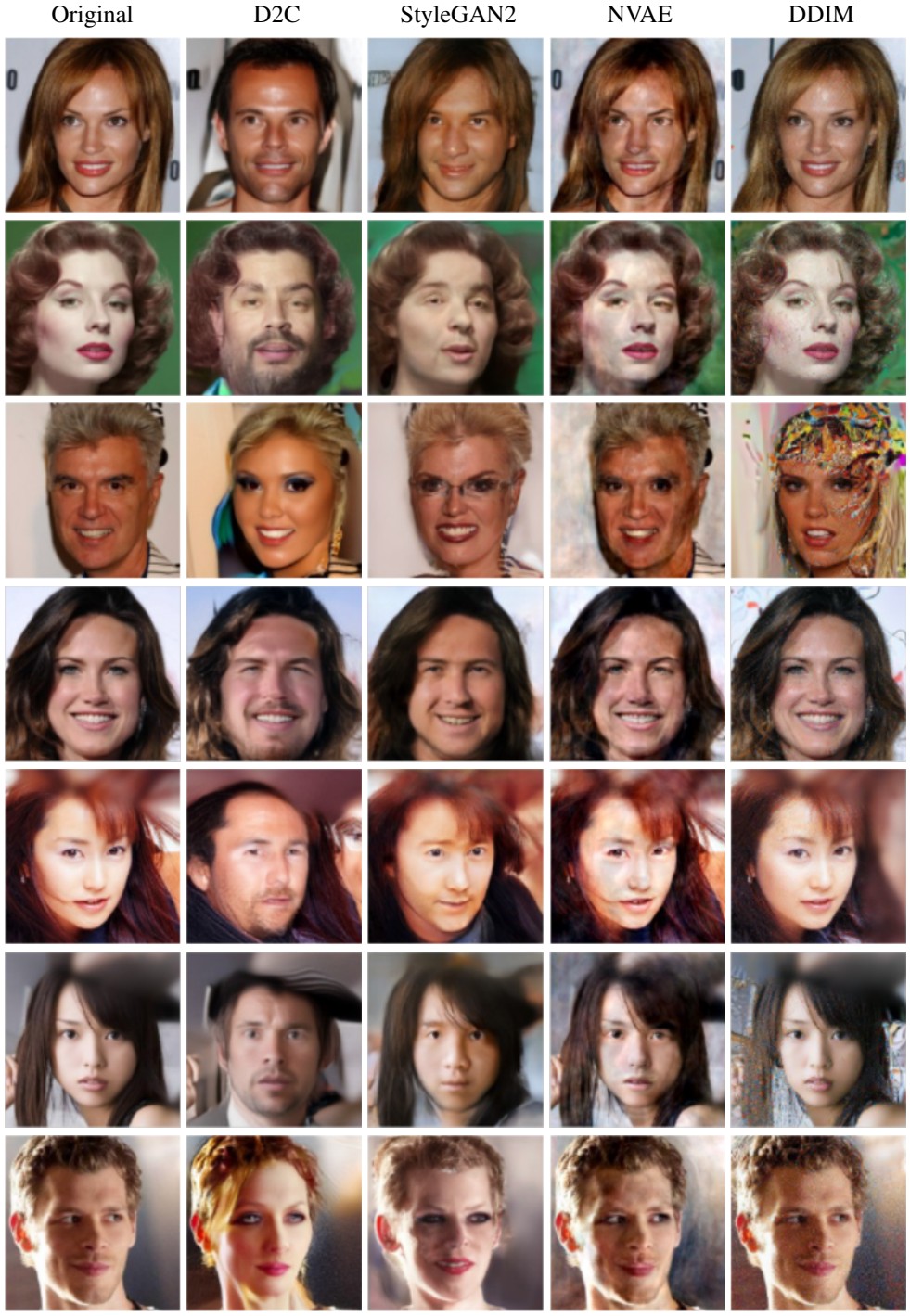

Figure 13: Image manipulation results for *gender*.

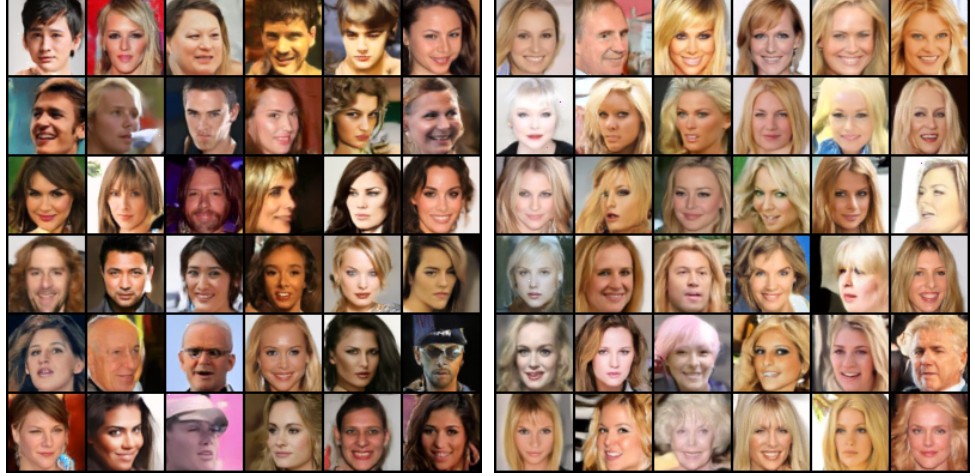

(a) Conditioned on *non-blond* label        (b) Conditioned on *blond* label

Figure 14: Conditional generation with D2C by learning from 100 labeled examples.

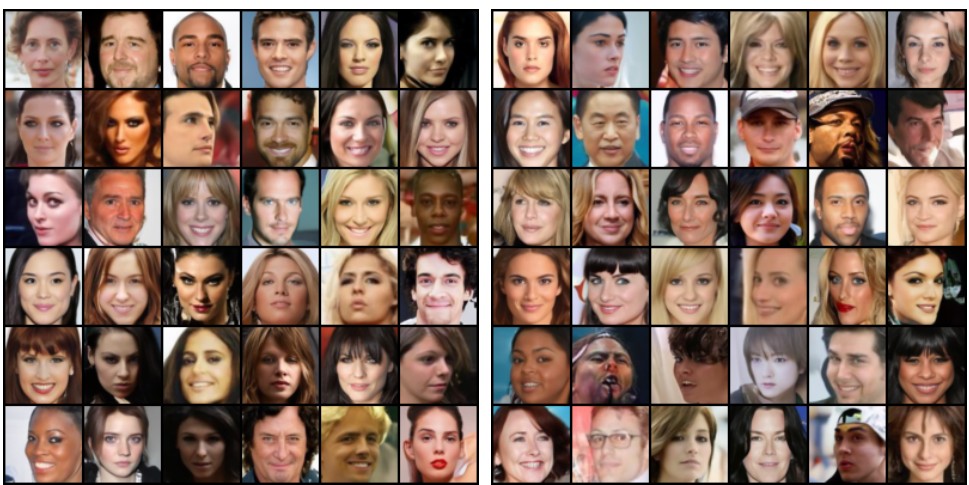

(a) Conditioned on *non-blond* label        (b) Conditioned on *blond* label

Figure 15: Conditional generation with DDIM by learning from 100 labeled examples.

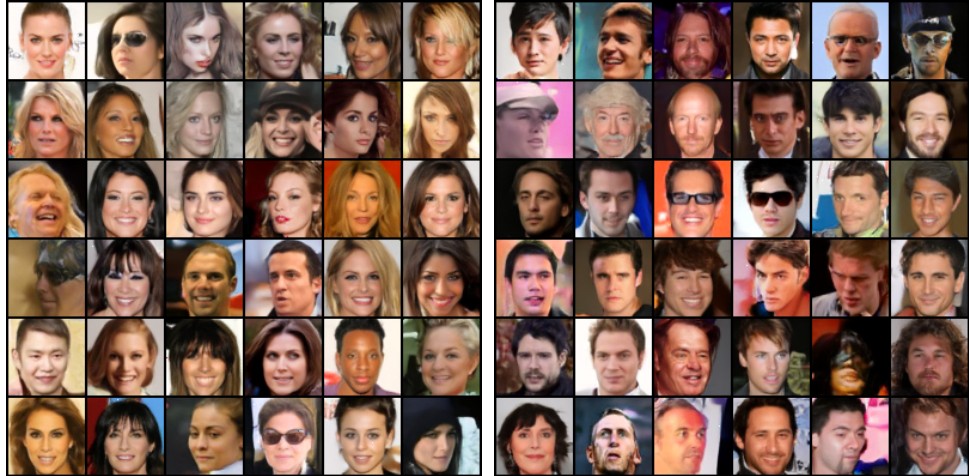

(a) Conditioned on *female* label  (b) Conditioned on *male* label

Figure 16: Conditional generation with D2C by learning from 100 labeled examples.

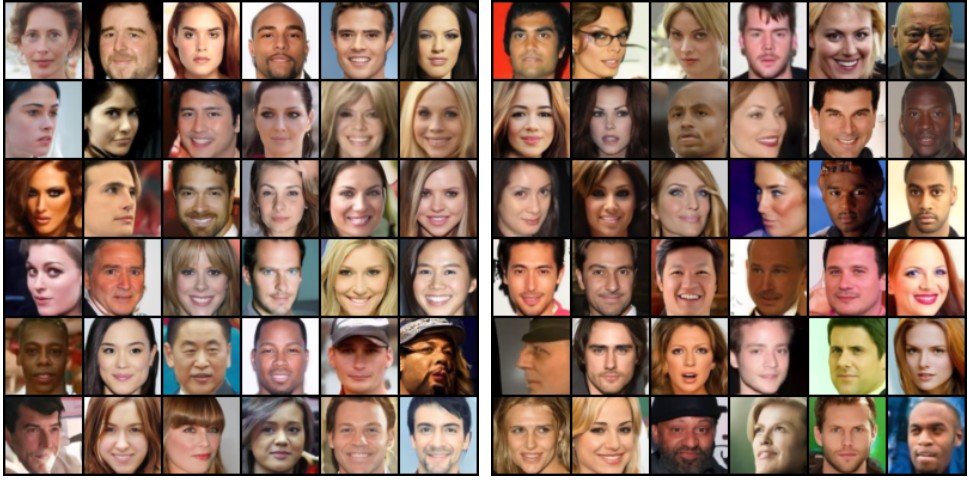

(a) Conditioned on *female* label  (b) Conditioned on *male* label

Figure 17: Conditional generation with DDIM by learning from 100 labeled examples.