# OpenReview forum: "D2C: Diffusion-Decoding Models for Few-Shot Conditional Generation"
_NeurIPS.cc/2021/Conference — NeurIPS 2021 Poster_

### Official Review · Reviewer_n2do · 2021-07-11

**Rating:** 6
**Confidence:** 5

**Summary:**

This paper proposes to learn a joint model consisting of a VAE and a latent space diffusion. The training process is augmented with a contrastive loss.

The authors show that this approach makes possible a few-shot generation of images with desired properties. To achieve this,  one needs to train a classifier in the latent space and after that sample latent codes with high classifier scores in any way.

Also, the case of semantic editing is evaluated. The manipulation is implemented using a Langevin-like approach in the latent space.

**Limitations And Societal Impact:**

No suggestions

**Main Review:**

1. Strengths.

    To the best of my knowledge, the proposed joint training of VAE and latent diffusion model is relatively novel, though the motivation for such an approach is pretty clear. Application of the contrastive loss to the diffusion model also has not been described earlier.  Also, I like the way the authors handle the problem of latent posterior mismatch, Theorem 2 is of particular interest.

2. Questions.

    a. First of all, I wonder which properties of diffusion make it a better candidate for latent posterior mismatch than, e.g. a normalizing flow? While training a diffusion model still contains a lot of heuristics (e.g. the usage of a so-called simple objective instead of a theoretically motivated variational bound is still a common choice [1] on par with synthetic hybrid objectives [2]), the training process of NF seems more stable. By the way, what weighting functions $w (\alpha\_i)$ (equation 2) were used during the training of D2C?

    b. Though the authors claim that the usage of contrastive loss was beneficial, (line 250, 269), there is no ablation study that proves that.

    c. Examples of image manipulation, presented in the paper, are not so impressive. To my mind, an example of gender editing, shown in the Appendix, is the most interesting one, while others are too simplistic. I think it could be more interesting to evaluate the presented approach on the 'smile' attribute or other attributes that require more complicated transformation that at the same time preserves the human identity.

    d. Figure 13b of the Appendix contains just a single image of a blond man. Is it due to the bias of labeled data? Or does this disproportion have another explanation?

3. Typos
    a. Equation 5: p -> p_\theta

    b. Line 138: x_0 -> z_0


4. Rating (pre-rebuttal)

    To my mind, this paper has demonstrated appealing results, but a more thorough evaluation of the proposed model's properties is required.

5. Updates (post-rebuttal)

   After reading other reviews and the authors' feedback, I retain my original assessment.

**Time Spent Reviewing:**

4

---

> ### Author Response · Authors · 2021-08-10
> **Response to Reviewer n2do**
>
> Thank you for your detailed review and thoughtful feedback. We address some of your comments below:
>
> **Q1: “I wonder which properties of diffusion make it a better candidate for latent posterior mismatch than, e.g. a normalizing flow?”**
>
> A: The property is relevant to Theorem 2. Intuitively, we want to make sure that for the latents that we use during generation, we should have already “seen” them during training (more precisely, we should have “seen” a “close enough” one with high probability).
>
> In normalizing flows, the latent distribution accessed during training is the pushforward of the training set with the flow model (similar to regular VAE encoders). Theorem 2 states that this distribution can be quite different from the one used for sampling, even though the KL divergence between them is small enough.
>
> From another angle, the dimensions of the latents (3k) is very high compared to the amount of data samples we have (50k for CIFAR). So even when two distributions can be quite different (in terms of support space), it would be nearly impossible to tell them apart with just 50k samples. As a result, there are many latent variables that we use during generation, but have not been “seen” during training; analogous to adversarial examples, samples from these latents are not expected to have good sample quality.
>
> Diffusion models and GANs do not have the above problem, since the latent distribution used for training and generation are the same by construction (and not made "close" by optimization). This property sets diffusion models apart from regular NFs even though DDIM itself can be viewed as a discretization of a probability flow ODE.
>
> **Q2: What weighting functions were used during the training of D2C?**
>
> A: The simple weights of 1 for all noise levels are used.
>
> **Q3: Additional image manipulation results are warranted.**
>
> A: Thank you for the comments. In the past week, we performed some preliminary experiments on the smile attribute. Although D2C added white teeth to the faces while keeping the identity, the samples do appear slightly unrealistic (see [https://pasteboard.co/Kf8bFR2.png](https://pasteboard.co/Kf8bFR2.png)), so we do not believe it would outperform StyleGAN2 in AMT evaluation at the moment.
>
> Nevertheless, we believe that this can be improved with a style-based generator, and will include these results in the final version.
>
> **Q4: “Figure 13b of the Appendix contains just a single image of a blond man.”**
>
> A: Thank you for pointing this out.
>
> To be precise, there are three (row 1 column 2, row 4 column 3 [with beard and long hair], row 5 column 6), but indeed the gender distribution is quite skewed in CelebA.
>
> We quote [1]: “Of the four groups (blond male, blond female, non-blond male, non-blond female), blond males are the smallest group, with only 1,387 examples out of 162,770 training examples, and they tend to be the worst group empirically”. Total blond haired people is around 24,415, so P(males | blond) in the CelebA dataset is around 5.6%.
>
> Surprisingly, this is very close to the ratio of blond males that D2C generates (2 or 3 out of 36), which suggests that D2C is reasonably faithful to the data distribution (for better or worse). Nevertheless, this means that we can encourage a “more balanced” data generation procedure by conditioning on both “blond” and “gender” features.
>
> To illustrate this point, we conditionally generate blond images with the same number of two genders in this figure [https://pasteboard.co/KfcPgWA.png](https://pasteboard.co/KfcPgWA.png), with top three rows being “blond female” and bottom three rows being “blond male”. While these are not entirely perfect, we note that this is only trained with 400 labeled data (male/female/blond/non-blond), so there could be some spurious correlation that is captured beyond the intended label.
>
>
>
> [1] Jones, E., Sagawa, S., Koh, P.W., Kumar, A. and Liang, P., 2020. Selective classification can magnify disparities across groups. arXiv preprint arXiv:2010.14134.

---

> > ### Comment · Reviewer_n2do · 2021-08-26
> > **Reviewer Response**
> >
> > I thank the authors for their feedback. The review has been updated based on the overall discussion.
> > I apologize for incorrectly counting the blond men in Figure 13b - that was my mistake.

---

> > > ### Author Response · Authors · 2021-09-03
> > > **Thank you for the comment.**
> > >
> > > Thank you for pointing out this issue about blond males in the samples, which gave us an opportunity to further investigate the "fairness" of the model. We will include these additional experimental results in the final version of the paper.

---

### Official Review · Reviewer_28ig · 2021-07-15

**Rating:** 6
**Confidence:** 4

**Summary:**

D2C proposes three main contributions to diffusion models, a recently popularized class of likelihood-based generative models based on parameterized Markov chains, particularly successful when viewed from a denoising score matching lens. First, this paper learns a prior for the *latent space* of an image VAE (NVAE), while past work mostly applied diffusion to the data space. Second, D2C adds a SimCLR-style contrastive loss to image latents in order to improve the quality of latent representations for classification. Third, binary attribute classifiers are learned conditioned on these latents and used to guide prior sampling with rejection sampling or Langevin dynamics. Experimental results qualitatively and quantitatively show some success in conditional image generation and manipulation.

**Limitations And Societal Impact:**

Yes.

**Main Review:**

This is an interesting paper with novel contributions in its hybridization of single modality contrastive representation learning, image generation and conditional guidance. Its findings on the success of diffusion in latent space should be exciting to researchers in the field. The writing is clear. In my view, it makes good contributions, but can be improved significantly in experimental methodology, breadth of experiments, and fair comparisons.

Reference [59], on Symbolic Music Generation, is more relevant than is given credit in the paper. [59] also develops a diffusion prior for VAEs, and this should be made clear.

Is the MoCo contrastive loss computed with augmentations to images or directly to latents? Does this apply during NVAE pre-training?

In 7.1, is it possible to measure the reconstruction error of D2C without L_D2? An ablation of L_C in terms of FID, MSE and Accuracy would also add quite a bit of insight, as the addition of the contrastive loss is a core contribution of the work. Is D2C FID’s worse than DDIM on CIFAR and fMoW due to the contrastive loss, the proposed diffusion in latent space, or other architectural details? Currently, these contributions are entangled, and their relative contributions are unclear to me. Clarifying this is one of my main concerns.

My other main concern is the limited breadth of conditional generation and manipulation tasks. Only four attributes are considered in the paper -- male/female, beard/no beard, lipstick/no lipstick, and blond/non-blond, all on face datasets. Does the method improve class conditional generation e.g. for CIFAR-10, or manipulations on fMoW?

Also, how do GAN models compare on attribute conditional image generation in this few-shot setting?

Additional unconditional generation baselines should be added to Table 3, such as StyleGAN2. These numbers should be able to be copied from their respective papers, and would help contextualize the results. It’s understandable that encoder-free approaches are excluded from Table 2 due to the reconstruction MSE metric, but even there, 76% accuracy on CIFAR-10 is far from the state of the art (e.g. Image GPT has higher than 90% without any contrastive loss).

# Minor comments
Why do conditional generations use rejection sampling, while image manipulations use langevin sampling from the reweighted density? It seems rejection sampling would become inefficient with higher class counts than the binary attribute settings considered in the paper.

The notation flips the usual time convention for diffusion models, where during generation time runs backward from T to 0. In your exposition, the alpha parameter is varied from 0 to 1 during latent generation. It’s OK since you define the convention, but is a bit confusing.

Alg 1 L2 should have q_\phi(z^1|x_i) not x

**Time Spent Reviewing:**

7

---

> ### Author Response · Authors · 2021-08-10
> **Response to Reviewer 28ig**
>
> Thank you for your detailed review and thoughtful feedback. We address some of your comments below:
>
> **Q1: “Reference [59], on Symbolic Music Generation, is more relevant than is given credit in the paper.”**
>
> A: Indeed, [59] learns a diffusion model over the aggregate posterior of a Music VAE. We will make this point clear in the final version.
>
> **Q2: “Is the MoCo contrastive loss computed with augmentations to images or directly to latents? Does this apply during NVAE pre-training?”**
>
> A: In D2C, the augmentations are performed over images (using standard augmentation techniques in self-supervised learning). This is not applied to NVAE pre-training where we follow the approach of the authors.
>
> **Q3: “In 7.1, is it possible to measure the reconstruction error of D2C without L_D2?”**
>
> A: We used a pre-trained MoCo-v2 model and trained a NVAE decoder to reconstruct the image. The reconstruction MSE per image was 58.20, significantly worse than NVAE (0.25) and D2C (0.76). The FID of the reconstructed images is 49, which is much higher than our methods (which has an reconstruction FID of around 1).
>
> We believe this could be relevant to the downsampling layers in ResNet; in the paper, we mentioned that we have tried ResNet encoders in D2C, which also led to much higher reconstruction errors (and worse FID).
>
> **Q4: “An ablation of L_C in terms of FID, MSE and Accuracy would also add quite a bit of insight, as the addition of the contrastive loss is a core contribution of the work.”**
>
> A: We performed an additional ablation study on this topic, where we train a D2 model (without the contrastive learning component), which applies a diffusion model over the latent variables. The experiment is identical to what Reviewer 1 asked for (although their focus is over the generative performance of the D2 model).
>
> This table shows the FID score of the generated images with a different number of diffusion steps.
>
> |        |       | CIFAR10 |       |       | CIFAR100 |        |
> |-------|-------|-------|-------|-------|-------|-------|
> | Steps | 10    | 50  | 100   | 10    | 50       | 100   |
> | D2      | 22.3 | 15.8 | 15.1  | 28.35 | 19.81    | 19.85 |
> | D2C   | 17.71 | 10.11| 10.15 | 23.16 | 14.62    | 14.46 |
>
> This table shows the MSE, FID and latent representation accuracy comparisons between D2, D2C and NVAE.
>
> |        |       | CIFAR10 |       |       | CIFAR100 |       |
> |-------|-------|---------|-------|-------|----------|-------|
> |        | FID   | MSE     | Acc   | FID   | MSE      | Acc   |
> | D2    | 15.1  | 0.24    | 40.6  | 19.85 | 0.48     | 17.89 |
> | D2C   | 10.15 | 0.76    | 76.02 | 14.62 | 0.44     | 42.75 |
> | NVAE  | 36.4  | 0.25    | 18.8  | 42.5  | 0.53     | 4.1   |
>
> Compared with the performance of NVAE (36.4 on CIFAR10 and 42.5 on CIFAR100), even D2 is significantly better. Additionally, D2C is even better than D2 in terms of unconditional generation performance.
>
> **Q5: “Is D2C FID’s worse than DDIM on CIFAR and fMoW due to the contrastive loss, the proposed diffusion in latent space, or other architectural details?”**
>
> A: We believe this to be the case of architecture issues and compute issues, as we adopted an NVAE architecture in order to compare with NVAE directly. The amount of compute in training could also be a factor here, as we spent roughly one-third of the compute training our model than the DDIM one. In the final section, we mentioned that adopting novel GAN architectures (such as StyleGAN2 and Gansformer) could be key to improved performance.
>
> **Q6: “... limited breadth of conditional generation and manipulation tasks. Does the method improve class conditional generation e.g. for CIFAR-10, or manipulations on fMoW?”**
>
> A: To address your concern, we performed an additional experiment on conditional generation on CIFAR-10. In this experiment, we train a classifier with 50 labels in each class. Then for each label, we evaluate the FID between the real images and generated images under that label, and take the average. The average FID for D2C/DDIM/NVAE is 32.57, 82.78, and 95.45 respectively (note that FIDs are expected to be higher because it is evaluated with 5k real samples per class instead of the commonly used 50k). This shows that D2C performs significantly better than DDIM and NVAE on the few-shot conditional generation task for CIFAR-10 as well. We will include additional results in the final version.
>
> **Q7: “how do GAN models compare on attribute conditional image generation in this few-shot setting?”**
>
> A: This setting is not directly applicable to GANs because of the need to find the correct latents for the inputs (GAN inversion). Granted, a similar approach to that of image manipulation can be adopted, but as GAN inversion can be quite slow (2 orders of magnitude slower than D2C), we did not have the resources in the past week to train a CIFAR-10/CIFAR-100/CelebA StyleGAN2 + inversion model to directly compare against our methods. However, given its success in image manipulation (e.g., gender), we believe that StyleGAN2 should perform this task reasonably well on face datasets.
>
> **Q8: “Image GPT has higher than 90% without any contrastive loss”**
>
> A: We would like to note that the model in Image GPT is trained over the ImageNet dataset, which is about 20x larger than the CIFAR10 dataset, so the two results are not directly comparable; the computing resources used are also much higher than ours. Nevertheless, it would be interesting to adopt such architectural improvements into D2C as future work.
>
> **Q9: “Why do conditional generations use rejection sampling?”**
>
> A: Indeed. Our goal in this paper, though, is to show that adding contrastive loss helps with the few-shot conditional generation task, for which rejection sampling is a simple and correct implementation. More efficient approximate implementations, such as classifier guidance, can be adopted as well [1]; we leave it as future work.
>
> **Q10: Notations.**
>
> Thank you for identifying the typo. We will fix this in the final version. The use of alpha notation abstracts away the exact diffusion generation procedure that is used; instead of a fixed T steps, we can choose different step schedules with DDIM.
>
> [1] Dhariwal, P. and Nichol, A., 2021. Diffusion models beat gans on image synthesis. arXiv preprint arXiv:2105.05233.

---

> > ### Comment · Reviewer_28ig · 2021-08-26
> > **Maintaining review**
> >
> > I appreciate the additional limited data CIFAR 10 experiment and D2 generation results. The latter are especially insightful and should be added to the paper.
> >
> > I still believe additional GAN manipulation baselines are important and push back on the criticism of the cost of latent inversion. Yes, inversion is expensive, but it only needs to be applied at test time, not during training. GAN inversion can produce quite compelling data restorations such as in http://proceedings.mlr.press/v139/daras21a.html. The Daras et al paper uses 50 - 1000 iterations for inversions (Sec 4.3), which is comparable to the cost of sampling from a diffusion model.
> >
> > At this time, I will maintain my score.

---

> > > ### Author Response · Authors · 2021-09-03
> > > **Sampling from diffusion model**
> > >
> > > Thank you for the comment. We will provide GAN results in the final version with an unconditional StyleGAN model on CIFAR-10.
> > >
> > > We would like to note that we only used 5 timesteps of the smaller latent diffusion model at test time, which is much faster compared to 50 - 1000 iterations of iterations through the large StyleGAN2 generator. As a result, we find that inversion + manipulation with D2C is much faster than StyleGAN2 (where it could take minutes for each image). In the final version of the paper, we will include more details about the procedure of this experiment and cite the related work that you mentioned.

---

### Official Review · Reviewer_Sq7M · 2021-07-16

**Rating:** 7
**Confidence:** 4

**Summary:**

The authors present Diffusion Decoding Models with Contrastive Representations (D2C) - a special VAE model with diffusion modeling in the latent space combined with contrastive learning based training of the inference network. The authors claim that D2C helps avoid the prior hole issue with VAEs due to diffusion modeling of the latent space, while contrastive learning helps learn meaningful representations allowing Few-Shot conditional generation. They further demonstrate that D2C outperforms NVAE and DDIM in few-shot conditional generation while remaining comparable to DDIM in unconditional generation.

**Ethics Review Area:**

["Discrimination / Bias / Fairness Concerns"]

**Limitations And Societal Impact:**

The authors adequately address the limitations and potential negative societal impact of their work.



**Main Review:**

- The paper applies 1) existing methods of diffusion models in the latent space of a VAE model, and 2) contrastive representation learning methods in the latent space of VAE.
- All sections of the paper are clearly written and easy to follow.
- D2C achieves strong results on few-shot conditional generation compared to a strong VAE baseline.
- In image manipulation experiments, while StyleGAN2 generates more "realistic looking images", they are poor at preserving details. D2C ,on the other hand, seems to preserve features but may produce distorted images (Figure 1 bottom image). So, there is a tradeoff between these two methods. However, D2C does perform qualitatively better manipulations than NVAE.
- It would be great to perform some analysis on the role of diffusion modeling in the latent space vs contrastive representation learning methods aka the two main modifications over a standard VAE. E.g. how good is a NVAE model when trained with an auxiliary contrastive representation learning objective? Such an analysis would make the contributions more significant for future works in terms of recognizing the importance of each component in latent variable generative models.

**Needs Ethics Review:**

Yes

**Time Spent Reviewing:**

3

---

> ### Author Response · Authors · 2021-08-10
> **Response to Reviewer Sq7M**
>
> Thank you for your detailed review and thoughtful feedback. We address some of your comments below:
>
> **Q1: “It would be great to perform some analysis on ... the two main modifications over a standard VAE”**
>
> A: We train a diffusion-denoising (D2) model (without the contrastive learning component). This table shows the FID score of the generated images with a different number of diffusion steps.
>
> |        |       | CIFAR10 |       |       | CIFAR100 |        |
> |-------|-------|-------|-------|-------|-------|-------|
> | Steps | 10    | 50  | 100   | 10    | 50       | 100   |
> | D2      | 22.3 | 15.8 | 15.1  | 28.35 | 19.81    | 19.85 |
> | D2C   | 17.71 | 10.11| 10.15 | 23.16 | 14.62    | 14.46 |
>
> Compared with the performance of NVAE (36.4 on CIFAR10 and 42.5 on CIFAR100), D2 is significantly better. Moreover, D2C is better than D2 in terms of unconditional generation performance.
>
> This table shows the MSE, FID and latent representation accuracy comparisons between D2, D2C, and NVAE.
>
> |        |       | CIFAR10 |       |       | CIFAR100 |       |
> |-------|-------|---------|-------|-------|----------|-------|
> |        | FID   | MSE     | Acc   | FID   | MSE      | Acc   |
> | D2    | 15.1  | 0.24    | 40.6  | 19.85 | 0.48     | 17.89 |
> | D2C   | 10.15 | 0.76    | 76.02 | 14.62 | 0.44     | 42.75 |
> | NVAE  | 36.4  | 0.25    | 18.8  | 42.5  | 0.53     | 4.1   |
>
> Here, the D2 has worse latent representation accuracy than D2C but better than NVAE.
>
>  We also attempted an experiment with NVAE + contrastive loss, but at the moment, we were unable to achieve satisfactory generation results (reconstruction MSE remains high). This is possibly due to the many regularizations needed for NVAE to work well, which could conflict with contrastive learning ([https://github.com/NVlabs/NVAE#known-issues](https://github.com/NVlabs/NVAE#known-issues)); D2 and D2C did not adopt these regularizations, just the NVAE architecture.
>
> We will continue to work on additional ablation experiments and place them in the final version.

---

> > ### Comment · Reviewer_Sq7M · 2021-08-29
> > **Thank you for the response.**
> >
> > I thank the authors for their response. The additional ablation results comparing D2, D2C, and NVAE seem to suggest that while contrastive learning has a significant impact on the performance, D2 is already better than NVAE by a big margin. As the authors pointed out, I think it will be very helpful to include NVAE + C ablation in the paper, or at least mentioning that such a direct combination of NVAE + C doesn't work out as simply as D2 + C (e.g. because of the regularizations, as the authors mentioned).

---

> > > ### Author Response · Authors · 2021-09-03
> > > **Thank you for the comment.**
> > >
> > > We will include these results in the final version of the paper and report the NVAE + C evaluation as well.

---

### Official Review · Reviewer_5hPh · 2021-07-21

**Rating:** 7
**Confidence:** 4

**Summary:**

The paper describes two methods: 1) how to incorporate self-supervised learning in VAE to have a meaningful encoded representation, 2) how to use diffusion model over later to bridge the gap between posterior and prior for unconditional generation, 3) how to use the given setup to do few-shot conditional generation. Empirically, authors show highly improved representation quality, which can be attributed to the incorporation of self-supervised learning. Additionally, experiments show improved FID scores for face datasets, but not for CIFAR. Authors don’t show results on ImageNet. The authors show improved FID scores for the conditional generation task. There is no FID comparison against StyleGAN2 (or any other GAN-based model), even if the authors claim to come very close in sample quality to such models. The qualitative depiction of image manipulation tasks is very impressive.

**Limitations And Societal Impact:**

The authors address the limitations and societal impact sufficiently.

**Main Review:**


The paper is a delight to read. All components are explained well, and the arguments are easy to follow. The contributions are novel and will be valuable to the rest of the community.

Pros:
1. The proposed approach seems to alleviate the prior hole problem for VAEs. However, it comes at the cost of deviating from optimizing the true lower bound on likelihood. FID comparison against GAN models will strengthen the paper.
2. The proposed approach is much faster at image manipulation tasks, which could be very valuable for many applications.

Cons:
1. Addition of self-supervised learning and diffusion model over latent is confounding. It would be great to study the effectiveness of diffusion model over latent as a solution to the prior hole problem in VAEs.
2. FID score comparison against GANs for image quality is warranted since authors claim to come close to GANs in sample quality.
3. Discussion about the strength/weakness of the model with varying data diversity would be useful since the FID scores don’t improve for non-face datasets.

**Time Spent Reviewing:**

6

---

> ### Author Response · Authors · 2021-08-10
> **Response to Reviewer 5hPh**
>
> Thank you for your detailed review and thoughtful feedback. We address some of your comments below:
>
> **Q1: The D2C objective deviates from optimizing the true lower bound on likelihood.**
>
> A: It has been observed that the evidence lower bound (ELBO) used for VAE objectives do not necessarily learn informative latent representations [1]. To encourage useful latents, one can introduce additional objective functions that maximize the mutual information between the latents and the observations [2]. The contrastive loss in D2C is based on InfoNCE, which is also a lower bound to mutual information [3]. By not exactly optimizing ELBO, D2C comes with the benefit of being able to learn more informative latent representations.
>
> **Q2: “It would be great to study the effectiveness of diffusion model over latent as a solution to the prior hole problem in VAEs.”**
>
> A: We performed an additional ablation study on this topic, where we train a D2 model (without the contrastive learning component), which applies a diffusion model over the latent variables.
>
> This table shows the FID score of the generated images with a different number of diffusion steps.
>
> |        |       | CIFAR10 |       |       | CIFAR100 |        |
> |-------|-------|-------|-------|-------|-------|-------|
> | Steps | 10    | 50  | 100   | 10    | 50       | 100   |
> | D2      | 22.3 | 15.8 | 15.1  | 28.35 | 19.81    | 19.85 |
> | D2C   | 17.71 | 10.11| 10.15 | 23.16 | 14.62    | 14.46 |
>
> Compared with the performance of NVAE (36.4 on CIFAR10 and 42.5 on CIFAR100), even D2 is significantly better. Additionally, D2C is even better than D2 in terms of unconditional generation performance.
>
> This table shows the MSE, FID and latent representation accuracy comparisons between D2, D2C and NVAE.
>
> |        |       | CIFAR10 |       |       | CIFAR100 |       |
> |-------|-------|---------|-------|-------|----------|-------|
> |        | FID   | MSE     | Acc   | FID   | MSE      | Acc   |
> | D2    | 15.1  | 0.24    | 40.6  | 19.85 | 0.48     | 17.89 |
> | D2C   | 10.15 | 0.76    | 76.02 | 14.62 | 0.44     | 42.75 |
> | NVAE  | 36.4  | 0.25    | 18.8  | 42.5  | 0.53     | 4.1   |
>
> Here, the D2 has worse latent representation accuracy than D2C but better than NVAE. These tables suggest that while adding a diffusion model over the latent space is beneficial (since D2 outperforms NVAE), adding the contrastive component may further improve performance.
>
> In Theorem 2, we also present an argument as to why diffusion models are fundamentally superior to other types of latent priors in terms of generative modeling.
>
> **Q3: “FID score comparison against GANs for image quality”**
>
> A: We will include the FID score comparison in the final version. While D2C does not outperform state-of-the-art StyleGANs, it outperforms most other GAN approaches (such as BigGAN, whose FIDs on unconditional CIFAR-10/100 are 18.64 and 22.19, respectively). Compute resources and model architecture also greatly affect the performance of a model (e.g., diffusion models were able to beat GANs with such improvements[4]), so we believe that D2C can also benefit from these improvements.
> We also compared with StyleGAN2 in terms of image manipulation (which could be hard to evaluate with automated metrics), showing ample promise for a VAE model to work well in this domain.
>
> **Q4: “Discussion about the strength/weakness of the model with varying data diversity would be useful”**
>
> A: We will include such a discussion in the final version. It appears that CIFAR-10 and CIFAR-100 are more complex than the face datasets, which may be verified with topological data analysis techniques [5].
>
> [1] Chen, X., Kingma, D.P., Salimans, T., Duan, Y., Dhariwal, P., Schulman, J., Sutskever, I. and Abbeel, P., 2016. Variational lossy autoencoder. arXiv preprint arXiv:1611.02731.
>
> [2] Zhao, S., Song, J. and Ermon, S., 2018. The information autoencoding family: A lagrangian perspective on latent variable generative models. arXiv preprint arXiv:1806.06514.
>
> [3] Poole, B., Ozair, S., Van Den Oord, A., Alemi, A. and Tucker, G., 2019, May. On variational bounds of mutual information. In International Conference on Machine Learning (pp. 5171-5180). PMLR.
>
> [4] Dhariwal, P. and Nichol, A., 2021. Diffusion models beat gans on image synthesis. arXiv preprint arXiv:2105.05233.
>
> [5] Khrulkov, V. and Oseledets, I., 2018, July. Geometry score: A method for comparing generative adversarial networks. In International Conference on Machine Learning (pp. 2621-2629). PMLR.

---

> > ### Comment · Reviewer_n2do · 2021-08-26
> > **Drop in Acc. for D2C on CIFAR-10**
> >
> > I would like to ask about the reported linear classification accuracy (Acc.) for D2C on CIFAR-10 data. Do you have any explanation as to why it is so low in comparison with D2 results (10.15 vs 40.6)? Note that for CIFAR-100, by contrast, accuracy is higher for the D2C model. Or is it just a typo, and the results are swapped?

---

> > > ### Author Response · Authors · 2021-08-26
> > > **Sorry for the typo**
> > >
> > > Sorry for creating the confusion. That was a typo created when we copied the table from the D2 FID results (the same 10.15 should have been the FID number for 100 steps). We fixed the accuracy to 76.02 (which is the number we reported in the submission).

---

### Review · Ethics_Reviewer_YrQ8 · 2021-08-11

**Recommendation:**

I recommend that the authors add a sentence or two acknowledging the first possible negative use case I identified above (the relevant scenario in the guidelines is #3). As mentioned above, as best I can tell this issue is not specifically tied to any innovation in their work; instead it is naturally associated with the problem of conditional generation of images when applied to images of humans. Therefore, I do not think an extensive discussion is needed; a short acknowledgement of the concern similar to the existing sentence acknowledging the deep fake concern is appropriate.

**Ethical Issues:**

Yes

**Ethics Review:**

This paper does raise ethical issues in my opinion. As best I can tell, these issues are entirely associated with the general topic of this paper, conditional generative models of high-dimensional images, as opposed to the specific algorithmic advances proposed in the paper. However, by advancing the state of the art in this sub-field, this work inherits the ethical issues of the broader problem.

In terms of the potential negative societal impacts raised by the [Neurips 2021 Ethics Guidelines](https://neurips.cc/public/EthicsGuidelines), the most obvious negative use-case that occurred to me that a future technology which relied on the advances described in this paper could be used to create an exclusionary online atmosphere or promote racially-biased standards of beauty. For example, imagine that a future technology where selfies can be modified using the diffusion-decoding models described in this work, and rather than conditioning on anodyne labels such as "blond" or "red lipstick", labels associated with protected characteristics were used. I believe this example would fall under scenarios #3 listed in the guidelines: *Raise human rights concerns. For example: could the technology be used to discriminate, exclude, or otherwise negatively impact people, including impacts on the provision of vital services, such as healthcare and education, or limit access to opportunities like employment? Please consult the Toronto Declaration for further details.* A methodological aspect of the study that exacerbates this issue is the use of humans (in this case via Amazon Mechanical Turk) to comparatively evaluate different models; since any biases of the evaluators could influence the evaluation metric used to guide progress on this task (and hence, product development).

Another possible issue is the use of these models to *Deceive people in ways that cause harm*, scenario #7 in the guidelines. The authors note this possibility in their final sentence: *Nevertheless, we note that our model have to be used properly in order to mitigate potential negative societal impacts, such as deep fakes.*

---

> ### Author Response · Authors · 2021-08-16
> **Thank you for the comment!**
>
> In the final version, we will add a paragraph acknowledging the negatives use cases that you have mentioned.

---

### Review · Ethics_Reviewer_vUkp · 2021-08-13

**Recommendation:** n/a

**Ethics Review:**

No obvious ethical issues.

---

### Author Response · Authors · 2021-08-10
**Summary of responses**

We would like to thank all reviewers for providing high quality reviews and insightful feedback. We are encouraged that reviewers think the contributions are “novel and valuable to the rest of the community” (R1), the paper is “clearly written and easy to follow” (R2), “exciting to researchers in the field” (R3), and “has demonstrated appealing results” (R4).

(We denote 5hPh, Sq7M, 28ig, n2do as R1, R2, R3, R4 respectively)

We address comments individually. Here, we summarize additional experiments we conducted in response to the comments. We will include additional studies (e.g. on face datasets and fMoW) in the final version.

- Ablation experiment on D2 (without contrastive) for CIFAR-10/100, with FID/Acc/MSE. (R1, R2, R3)
- Few-shot class conditional generation on CIFAR-10 (R3).
- Reconstruction error/generation with contrastive self-supervised features (R3).
- Additional image manipulation results based on "smile" attribute (R4).
- Balanced conditional generation over the “blond” attribute (R4).

We will include these additional experiments and ablation studies in the final version.

---

### Decision · Program_Chairs · 2021-09-27

**Decision:**

Accept (Poster)

**Comment:**

The reviewers unanimously believe that the paper should be accepted. The paper focuses on an important problem and some interesting ideas such as guided diffusion and the use of self-supervised representations has been proposed. There are complaints about the quality of the baselines and the rigorousness of the evaluation. The highlighted results do show some artifacts, and the proposed approach sounds like a combination of several disjoint technique. Hence, I recommend acceptance as a poster.